# Continuous Doubly Constrained Batch Reinforcement Learning

**Rasool Fakoor[1], Jonas Mueller[1], Kavosh Asadi[1], Pratik Chaudhari[1,2], Alexander J. Smola[1]**
[1]Amazon Web Services, [2]University of Pennsylvania
`fakoor@amazon.com`

## Abstract

Reliant on too many experiments to learn good actions, current Reinforcement Learning (RL) algorithms have limited applicability in real-world settings, which can be too expensive to allow exploration. We propose an algorithm for batch RL, where effective policies are learned using only a fixed offline dataset instead of online interactions with the environment. The limited data in batch RL produces inherent uncertainty in value estimates of states/actions that were insufficiently represented in the training data. This leads to particularly severe extrapolation when our candidate policies diverge from one that generated the data. We propose to mitigate this issue via two straightforward penalties: a policy-constraint to reduce this divergence and a value-constraint that discourages overly optimistic estimates. Over a comprehensive set of 32 continuous-action batch RL benchmarks, our approach compares favorably to state-of-the-art methods, regardless of how the offline data were collected.

## 1 Introduction

Deep RL algorithms have demonstrated impressive performance in simulable digital environments like video games [41, 54, 55]. In these settings, the agent can execute different policies and observe their performance. Barring a few examples [37], advancements have not translated quite as well to real-world environments, where it is typically infeasible to experience millions of environmental interactions [11]. Moreover, in presence of an acceptable heuristic, it is inappropriate to deploy an agent that learns from scratch hoping that it may eventually outperform the heuristic after sufficient experimentation.

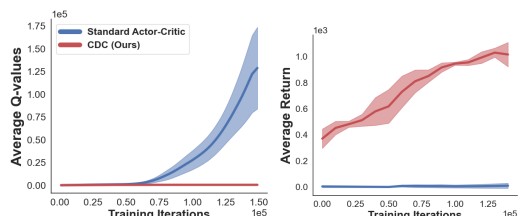

Figure 1: **Batch RL with CDC vs. No CDC**. Left: Standard actor-critic overestimates $Q$-values whereas CDC estimates are well controlled. Right: Wild overestimation leads to worse-performing policies whereas CDC performs well.

The setting of *batch* or *offline* RL instead offers a more pertinent framework to learn performant policies for real-world applications [34, 57]. Batch RL is widely applicable because this setting does not require that: a proposed policy be tested through real environment interactions, or that data be collected under a particular policy. Instead, the agent only has access to a fixed dataset $\mathcal{D}$ collected through actions taken according to some unknown *behavior* policy $\pi_{\mathrm{b}}$. The main challenge in this setting is that data may only span a small subset of the possible state-action pairs. Worst yet, the agent cannot observe the effects of novel out-of-distribution (OOD) state-action combinations that, by definition, are not present in $\mathcal{D}$.

A key challenge stems from the inherent uncertainty when learning from limited data [28, 36]. Failure to account for this can lead to wild extrapolation [17, 29] and over/under-estimation bias in value

35th Conference on Neural Information Processing Systems (NeurIPS 2021).

estimates [22, 23, 32, 58]. This is a systemic problem that is exacerbated for out-of-distribution (OOD) state-actions where data is scarce. Standard temporal difference updates to $Q$-values rely on the Bellman optimality operator which implies upwardly-extrapolated estimates tend to dominate these updates. As $Q$-values are updated with overestimated targets, they become upwardly biased even for state-actions well-represented in $\mathcal{D}$. In turn, this can further increase the upper limit of the extrapolation errors at OOD state-actions, which forms a vicious cycle of extrapolation-inflated overestimation (*extra-overestimation* for short) shown in Figure 1. This extra-overestimation is much more severe than the usual overestimation bias encountered in online RL [22, 58]. As such, we critically need to constrain value estimates whenever they lead to situations that look potentially 'too good to be true', in particular when they occur where a policy might exploit them.

Likewise, naive exploration can lead to policies that diverge significantly from $\pi_b$. This, in turn, leads to even greater estimation error since we have very little data in this un(der)-explored space. Note that this is not a reason for particular concern in online RL: after all, once we are done exploring a region of the space that turns out to be less promising than we thought, we simply update the value function and stop visiting or visit rarely. Not so in batch RL where we cannot adjust our policy based on observing its actual effects in the environment. These issues are exacerbated for applications with a large number of possible states and actions, such as the continuous settings considered in this work. Since there is no opportunity to try out a proposed policy in batch RL, learning must remain appropriately conservative for the policy to have reasonable effects when it is later actually deployed. Standard regularization techniques are leveraged in supervised learning to address such ill-specified estimation problems, and have been employed in the RL setting as well [12, 50, 62].

This paper adapts standard off-policy actor-critic RL to the batch setting by adding a simple pair of regularizers. In particular, our main contribution is to introduce two novel batch-RL regularizers: The first regularizer combats the extra-overestimation bias in regions that are out-of-distribution. The second regularizer is designed to hedge against the adverse effects of policy updates that severely diverge from $\pi_b(a|s)$. The resultant method, *Continuous Doubly Constrained Batch RL* (CDC) exhibits state-of-the-art performance across 32 continuous control tasks from the D4RL benchmark [14] demonstrating the usefulness of our regularizers for batch RL.

## 2   Background

Consider an infinite-horizon Markov Decision Process (MDP) [48], $(S, A, T, r, \mu_0, \gamma)$. Here $S$ is the state space, $A \subset \mathbb{R}^d$ is a (continuous) action space, $T : S \times A \times S \to \mathbb{R}_+$ encodes transition probabilities of the MDP, $\mu_0$ denotes the initial state distribution, $r(s, a)$ is the instantaneous reward obtained by taking action $a \in A$ in state $s \in S$, and $\gamma \in [0, 1]$ is a *discount factor* for future rewards.

Given a stochastic policy $\pi(a|s)$, the sum of discounted rewards generated by taking a series of actions $a_t \sim \pi(\cdot|s_t)$ corresponds to the *return* $R_t^\pi = \sum_{i=t}^{\infty} \gamma^{i-t} r(s_i, a_i)$ achieved under policy $\pi$. The *action-value function* (Q-value for short) corresponding to $\pi$, $Q^\pi(s, a)$, is defined as the expected return starting at state $s$, taking $a$, and acting according to $\pi$ thereafter, $Q^\pi(s, a) = \mathbb{E}_{s_t \sim T, a_t \sim \pi} \left[ \sum_{t=0}^{\infty} \gamma^t r_t \mid (s_0, a_0) = (s, a) \right]$. $Q^\pi(s, a)$ obeys the Bellman equation [5]:

$$Q^\pi(s, a) = r(s, a) + \gamma \mathbb{E}_{s' \sim T(\cdot|s,a), a' \sim \pi(\cdot|s')} \left[ Q^\pi(s', a') \right] \tag{1}$$

Unlike in online RL, no interactions with the environment is allowed here, so the agent does not have the luxury of exploration. $\mathcal{D}$ is previously collected via actions taken according to some unknown *behavior policy* $\pi_b(a|s)$. In this work, we assume $\mathcal{D}$ consists of 1-step transition: $\{(s_i, a_i, r_i, s_i')\}_{i=1}^n$ where no further sample collection is permitted. In particular, our method, like [60, 63], only needs a dataset consisting of a single-step transitions and does not require complete episode trajectories. This is valuable, for instance, whenever data privacy and sharing restrictions prevent the use of the latter [34]. It is also useful when combining data from sources where the interaction is still in progress, e.g. from ongoing user interactions.

We aim to learn an *optimal* policy $\pi^*$ that maximizes the expected return, denoting the corresponding Q-values for this policy as $Q^* = Q^{\pi^*}$. $Q^*$ is the fixed point of the Bellman *optimality* operator [5]: $\mathcal{T}Q^*(s, a) = r(s, a) + \gamma \mathbb{E}_{s' \sim T(\cdot|s,a)} \left[ \max_{a'} Q^*(s', a') \right]$. One way to learn $\pi^*$ is via actor-critic methods [27], with policy $\pi_\phi$ and Q-value $Q_\theta$, parametrized by $\phi$ and $\theta$ respectively.

Learning good policies becomes far more difficult in batch RL as it depends on the quality/quantity of available data. Moreover, for continuous control the set of possible actions is infinite, making it

nontrivial to find the optimal action even for online RL. One option is to approximate the maximization above by only considering finitely many actions sampled from some $\pi$. This leads to the Expected Max-Q (EMaQ) operator of Ghasemipour et al. [18]:

$$\overline{\mathcal{T}}Q(s,a) := r(s,a) + \gamma \mathbb{E}_{s' \sim T(\cdot|s,a)} \left[ \max_{\{a_k'\}} Q(s', a_k') \right]. \tag{2}$$

Here $a_k' \sim \pi_\phi(\cdot|s')$ for $k = 1, ..., N$, i.e. the candidate actions are drawn IID from the current (stochastic) policy rather than over all possible actions. When drawing only a single sample from $\pi_\phi$, this reduces to the standard Bellman operator (in expectation). Conversely, when $N \to \infty$ and $\pi_\phi$ has support over $A$, this turns into the Bellman optimality operator. We learn $Q$ by minimizing the standard 1-step temporal difference (TD) error. That is, we update at iteration $t$

$$\theta_t \leftarrow \underset{\theta}{\operatorname{argmin}} \, \mathbb{E}_{(s,a) \sim \mathcal{D}} \left[ \left( Q_\theta(s,a) - \overline{\mathcal{T}}Q_{\theta_{t-1}}(s,a) \right)^2 \right] \tag{3}$$

Throughout, the notation $\mathbb{E}_{(s,a) \sim \mathcal{D}}$ denotes an *empirical* expectation over dataset $\mathcal{D}$, whereas expectations with respect to $\pi$ are taken over the true underlying distribution corresponding to policy $\pi$. Next, we update the policy by increasing the likelihood of actions with higher Q-values:

$$\phi_t \leftarrow \underset{\phi}{\operatorname{argmax}} \, \mathbb{E}_{s \sim \mathcal{D}, \hat{a} \sim \pi_\phi(\cdot|s)} \left[ Q_{\theta_t}(s, \hat{a}) \right] \tag{4}$$

using off-policy gradient-based updates [53]. Depending on the context, we omit $t$ from $Q_{\theta_t}$ and $\pi_{\phi_t}$.

## 2.1 Extrapolation-Inflated Overestimation

When our Q-values are estimated via function approximation[1] (with parameters $\theta$), the Q-update can be erroneous and noisy [58]. Let $Q_{\theta_t}(s,a)$ denote the estimates of true underlying $Q^*(s,a)$ values at iteration $t$ of a batch RL algorithm that iterates steps (3) and (4), with $\pi_{\phi_t}$ denoting the policy that maximizes $Q_{\theta_t}$. For a proper learning method, we might hope that the *estimation error*, ER $:= Q_{\theta_t}(s,a) - Q^*(s,a)$, has expected value $= 0$ and variance $\sigma > 0$ for particular states/actions (the expectation here is over the sampling variability in the dataset $\mathcal{D}$ and stochastic updates in our batch RL algorithm). However even in this desirable scenario, Jensen's inequality nonetheless implies there will be *overestimation error* OE $:= \mathbb{E}[\max_a Q_{\theta_t}(s,a)] - \max_a Q^*(a,s) \geq 0$ for the actions currently favored by $\pi_{\phi_t}$. Here the expectation is over the randomness of the underlying dataset $\mathcal{D}$ and the learning algorithm. OE will be strictly positive when the estimation errors are weakly correlated and will grow with the ER-variance $\sigma$ [32, 58]. Under the Bellman optimality or EMaQ operator, these inflated estimates are used as target values in the next Q-update in (3), which thus produces a $Q_{\theta_{t+1}}(s,a)$ estimate that suffers from *overestimation bias*, meaning it is expected to exceed the true Q value even if this was not the case for initial estimate $Q_{\theta_t}(s,a)$ [13, 16, 22, 23, 30, 33, 58].

In continuous batch RL, ER may have far greater variance (larger $\sigma$) for OOD states/actions poorly represented in the dataset $\mathcal{D}$, as our function approximator $Q_{\theta_t}$ may wildly extrapolate in these data-scarce regions [25, 34]. This in turn implies the updated policy $\pi_{\phi_t}$ will likely differ significantly from $\pi_b$ and favor some action $\hat{a} = \operatorname{argmax}_a Q_{\theta_t}(s,a)$ that is OOD [17]. The estimated value of this OOD action subsequently becomes the target in the Q-update (3), and its OE will now be more severe due to the larger $\sigma$ [7]. Even though we only apply these Q-updates to non-OOD $(s,a) \in \mathcal{D}$ whose ER may be initially smaller, the severely overestimated target values can induce increased overestimation bias in $Q_{\theta_{t+1}}(s,a)$ for $(s,a) \in \mathcal{D}$. In a vicious cycle, the increase in $Q_{\theta_{t+1}}(s,a)$ for $(s,a) \in \mathcal{D}$ can cause extrapolated $Q_{\theta_{t+1}}$ estimates to also grow for OOD actions (as there is no data to ground these OOD estimates), such that overestimation at $s, a \in \mathcal{D}$ is further amplified through additional temporal difference updates. After many iterative updates, this *extra-overestimation* can eventually lead to the disturbing explosion of value estimates seen in Figure 1.

Several strategies address overestimation [13, 16, 22, 23, 30, 33]. Fujimoto et al. [17] proposed a straightforward convex combination of the extremes of an estimated distribution over plausible Q values. Given a set of estimates $Q_{\theta_j}$ for $j = 1, ...M$, they combine both the maximum and the minimum value for a given $(s,a)$ pair:

$$\overline{Q}_\theta(s,a) = \nu \min_j Q_{\theta_j}(s,a) + (1-\nu) \max_j Q_{\theta_j}(s,a) \tag{5}$$

---

[1]While overestimation bias has been mainly studied in regard to function approximation error [13, 16, 23, 33, 58], Hasselt [22] shows that overestimation can also arise in tabular MDPs due to noise in the environment.

Here $\nu \in (0, 1)$ determines how conservative we wish to be, and the $\min/\max$ are taken across $M$ Q-networks that only differ in their weight-initialization but are otherwise (independently) estimated. For larger $\nu > 0.5$, $\overline{Q}$ may be viewed as a *lower confidence bound* for $Q^*$ where the epistemic uncertainty in Q estimates is captured via an ensemble of deep Q-networks [9].

## 3 Methods

Our previous discussion of extra-overestimation suggests two key sources of potential error in batch RL. Firstly, a policy learned by our algorithm might be too different from the behavior policy, which can lead to risky actions whose effects are impossible to glean from the limited data. To address this, we propose to add an *exploration-penalty* in policy updates that reduces the divergence between our learned policy $\pi_\phi$ and the policy $\pi_b$ that generated the data. Secondly, we must restrict overestimation in Q-values, albeit only where it matters, that is, only when this leads to a policy exploiting overly optimistic estimates. As such, we only need to penalize suspiciously large Q-values for actions potentially selected by our candidate policy $\pi_\phi$ (e.g. if their estimated Q-value greatly exceeds the Q-value of actually observed actions).

### 3.1 Q-Value Regularization

While sequential interaction with the environment is a strong requirement that limits the practical applicability of online RL (and leads to other issues like exploration vs. exploitation), it has one critical benefit: although unreliable extrapolation of Q-estimates beyond the previous observations happens during training, it is naturally corrected through further interaction with the environment. OOD state-actions with wildly overestimated values are in fact likely to be explored in subsequent updates, and their values then corrected after observing their actual effect.

In contrast, extra-overestimation is a far more severe issue in batch RL, where we must be confident in the reliability of our learned policy before it is deployed. The issue can lead to completely useless Q-estimates. The policies corresponding to these wildly extrapolated Q-functions will perform poorly, pursuing risky actions whose true effects cannot be known based on the limited data in $\mathcal{D}$ (Figure 1 shows an example of how extra-overestimation can lead to the disturbing explosion of Q-value estimates).

To mitigate the key issue of extra-overestimation in $Q_\theta(s, a)$, we consider three particular aspects:

- An overall shift in Q-value is less important. A change from, say $Q_\theta(s, a)$ to $Q_\theta(s, a) + c(s)$ changes nothing about which action we might want to pick. As such, we only penalize the *relative* shift between $Q$-values.

- An overestimation of $Q_\theta(s, \hat{a})$ which still satisfies $Q_\theta(s, \hat{a}) \ll Q_\theta(s, a)$ for well-established $a, s \in \mathcal{D}$ will not change behavior and does not require penalization.

- Lastly, overestimation only matters if our policy is capable of discovering and exploiting it.

We use these three aspects to design a penalty for Q-value updates to be more pessimistic [7, 25].

$$\Delta(s, a) := \left[ \max_{\hat{a} \in \{a_1, \dots a_N\} \sim \pi_\phi(.|s)} Q_\theta(s, \hat{a}) - Q_\theta(s, a) \right]_+^2 \tag{6}$$

where $s, a \in D$. We can see that the first requirement is easily satisfied, since we only compare differences $Q_\theta(s, \hat{a}) - Q_\theta(s, a)$ for different actions, given the same state $s$. The second aspect is addressed by taking the maximum between 0 and $Q_\theta(s, \hat{a}) - Q_\theta(s, a)$. As such, we do not penalize optimism when it does not rise to the level where it would effect a change in behavior. Lastly, taking the maximum over actions drawn from the $\pi$ rather than from the maximum over all possible actions ensures that we only penalize when the overestimation would have observable consequences. As such, we limit ourselves to a rather narrow set of cases. As a result, we add this penalty to the Q-update:

$$\theta_t \leftarrow \underset{\theta}{\operatorname{argmin}} \, \mathbb{E}_{(s,a)\sim\mathcal{D}} \left[ \left( Q_\theta(s, a) - \overline{\mathcal{T}} Q_{\theta_{t-1}}(s, a) \right)^2 + \eta \cdot \Delta(s, a) \right] \tag{7}$$

**Anatomy of the extra-overestimation penalty $\Delta$.** Our proposed $\Delta$ penalty in (6) mitigates extra-overestimation bias by hindering the learned Q-value from wildly extrapolating large values for OOD

state-actions. Estimated values of actions previously never seen in (known) state $s \in \mathcal{D}$ are instead encouraged to not significantly exceed the values of the actions $a$ whose effects we have seen at $s$. Note that the temporal difference update and the extra-overestimation penalty $\Delta$ in (7) are both framed on a common scale as a squared difference between two Q-functions.

How $\Delta$ affects $\theta$ becomes evident through its derivative:

$$\boldsymbol{\nabla}_\theta \Delta(s,a) = \begin{cases} \left( \boldsymbol{\nabla}_\theta Q_\theta(s,\hat{a}) - \boldsymbol{\nabla}_\theta Q_\theta(s,a) \right) \varepsilon & \text{if } \varepsilon > 0 \\ 0 & \text{otherwise} \end{cases} \tag{8}$$

Here $\hat{a} := \arg\max_{\{\hat{a}_k\}_{k=1}^N} Q_\theta(s, \hat{a}_k)$ again taken over $N$ actions sampled from our current policy $\pi$, and $\varepsilon := Q_\theta(s,\hat{a}) - Q_\theta(s,a)$. $\Delta$ only affects certain temporal-differences where Q-values of (possibly OOD) state-actions have higher values than the $(s,a) \in \mathcal{D}$. In this case, $\Delta$ not only reduces $Q_\theta(s,\hat{a})$ by an amount proportional to $\varepsilon$, but this penalty also increases the value of the previously-observed action $Q_\theta(s,a)$ to the same degree. $\Delta$ thus results in a value network that favors previously observed actions. We will generally want to choose a large conservative value of $\eta$ in applications where we know either: that the behavior policy was of high-quality (since its chosen actions should then be highly valued), or that only a tiny fraction of the possible state-action space is represented in $\mathcal{D}$, perhaps due to small sample-size or a restricted behavior policy (since there may be severe extrapolation error).

## 3.2 Policy Regularization

In batch RL, the available offline data $\mathcal{D}$ can have varying quality depending on the behavior policy $\pi_b$ used to collect the data. Since trying out actions is not possible in batch settings, our policy network is instead updated to favor not only actions with the highest estimated Q-value but also the actions observed in $\mathcal{D}$ (whose effects we can be more certain of). Thus we introduce an *exploration penalty* to regularize the policy update step: $\phi \leftarrow \arg\max_\phi \mathbb{E}_{s \sim \mathcal{D}, \hat{a} \sim \pi_\phi(\cdot|s)} \left[ Q_\theta(s,\hat{a}) \right] - \lambda \cdot \mathbb{D}(\pi_b, \pi_\phi)$.

In principle, various $f$-divergences [10] or Integral Probability Metrics [42] could employed in $\mathbb{D}(\cdot, \cdot)$. In practice, we limit our choice to quantities that do not require estimating the behavior policy $\pi_b$. This leaves us with the reverse KL-divergence and IPMs in Hilbert Space [2]. If we further restrict ourselves to distances that do not require sampling from $\pi_\phi$, then only the reverse KL-divergence remains. We thus estimate

$$\text{KL}(\pi_b, \pi_\phi) = \mathbb{E}_{a \sim \pi_b(\cdot|s)}[\log \pi_b(a|s)] - \mathbb{E}_{a \sim \pi_b(\cdot|s)}[\log \pi_\phi(a|s)] \tag{9}$$

$$\propto -\mathbb{E}_{a \sim \pi_b(\cdot|s)}[\log \pi_\phi(a|s)] \approx -\frac{1}{m} \sum_{i=1}^m \log \pi_\phi(a_i|s) \tag{10}$$

whenever $a_i \sim \pi_b(\cdot|s)$. This is exactly what happens in batch RL where we have plenty of data drawn from the behavior policy, albeit no access to its explicit functional form. Note the first entropy term in (9) can be ignored when we aim to minimize the estimated KL in terms of $\pi_\phi$ (as will be done in our exploration penalty). Using (10), we can efficiently minimize an estimated reverse KL divergence without having to know/estimate $\pi_b$ or sample from $\pi_\phi$.

**Lemma 1** $\arg\max\limits_{\pi_\phi} \mathbb{E}_{s,a \sim \pi_\phi}[Q_\theta(s,a)] - \lambda \cdot \mathbb{E}_s \left[ \mathbb{D}\big(\pi_\phi(\cdot|s), \pi_b(\cdot|s)\big) \right]$ *is given by*

$\pi_\phi(s|a) = \dfrac{\pi_b(a|s)}{Z} \exp \left( \dfrac{Q_\theta(s,a)}{\lambda} \right)$   *if $\mathbb{D}$ is the forward KL divergence* $= \text{KL}(\pi_\phi(\cdot|s), \pi_b(\cdot|s))$

$\pi_\phi(s|a) = \dfrac{\pi_b(a|s)}{Z - Q_\theta(s,a)/\lambda}$   *if $\mathbb{D}$ is the reverse KL divergence* $= \text{KL}(\pi_b(\cdot|s), \pi_\phi(\cdot|s))$

where $Z \in \mathbb{R}$ is a normalizing constant in each case. Lemma 1 shows that using either forward or reverse KL-divergence as an objective, we recover $\pi_\phi = \pi_b$ in the limit of $\lambda \to \infty$. This is to be expected. After all, in this case we use the distance in distributions (thus policies) as our only criterion, and we prefer reverse KL to avoid having to estimate $\pi_b$. CDC thus employs the following policy-update (where the reverse KL is expressed as a log-likelihood as in (10))

$$\phi \leftarrow \arg\max_\phi \mathbb{E}_{s \sim \mathcal{D}, \hat{a} \sim \pi_\phi(\cdot|s)} \left[ Q_\theta(s,\hat{a}) \right] + \lambda \cdot \mathbb{E}_{(s,a) \sim \mathcal{D}} \left[ \log \pi_\phi(a|s) \right] \tag{11}$$

The exploration penalty helps ensure our learned $\pi_\phi$ is not significantly worse than $\pi_b$, which is far from guaranteed in batch settings without ever testing an action. If the data were collected by a fairly random (subpar) behavior policy, then this penalty (in expectation) acts similarly to a maximum-entropy term. The addition of such terms to similar policy-objectives has been shown to boost performance in RL methods like *soft actor-critic* [21].

Note that our penalization of exploration stands in direct contrast to online RL methods that specifically incentivize exploration [4, 44]. In the batch RL, exploration is extremely dangerous as it will only take place during deployment when a policy is no longer being updated in response to the effect of its actions. Constraining policy-updates around an existing data-generating policy has also been demonstrated as a reliable way to at least obtain an improved policy in both batch [17, 63] and online [51] settings. Even moderate policy-improvement can often be extremely valuable (the optimal policy may be too much ask for with data of limited size or coverage of the possible state-actions). *Reliable* improvement is crucial in batch settings as we cannot first test out our new policy.

**Remark 1 (Behavioral cloning occurs as $\lambda \to \infty$)** *Regularized policy updates with strong regularization (large $\lambda$) is in the limit imitation learning. In fact, this is the well-known likelihood based behavioral cloning algorithm used by [47].*

If the original behavior policy $\pi_b^*$ was optimal (e.g. demonstration by a human-expert), then behavioral cloning should be utilized for learning from $\mathcal{D}$ [43]. However in practice, data are often collected from a subpar policy that we wish to improve upon via batch RL rather than simple imitation learning.

### 3.3 CDC Algorithm

Furnished with the tools for Q-value and policy regularization proposed in previous sections, we introduce CDC in Algorithm 1. CDC utilizes an actor-critic framework [27] for continuous actions with stochastic policy $\pi_\phi$ and Q-value $Q_\theta$, parameterized by $\phi$ and $\theta$ respectively. Our major additions to that $\Delta$ penalty that mitigates overestimation bias by reducing wild extrapolation in value estimates and the *exploration penalty* ($\log \pi_\phi$) that discourages the estimated policy from straying to OOD state-actions very different from those whose effects we have observed in $\mathcal{D}$.

Although the particular form of CDC presented in Algorithm 1 optimizes a stochastic policy with the off-policy updates of [53] and temporal difference value-updates using (3), we emphasize that the general idea behind CDC can be utilized with other forms of actor-critic updates such as those considered by [12, 16, 21]. In practice, CDC estimates expectations of quanti-

---

**Algorithm 1** Continuous Doubly Constrained Batch RL

1: Initialize policy $\pi_\phi$ and Qs: $\{Q_{\theta_j}\}_{j=1}^M$
2: Initialize Target Qs: $\{Q_{\theta_j'} : \theta_j' \leftarrow \theta_j\}_{j=1}^M$
3: **for** $t$ in $\{1, \ldots, T\}$ **do**
4: $\quad$ Sample $\mathcal{B} = \{(s, a, r, s')\} \sim \mathcal{D}$
5: $\quad$ For each $s, s' \in \mathcal{B}$: sample $N$ actions $\{\hat{a}_k\}_{k=1}^N \sim \pi_\phi(\cdot|s), \{a_k'\}_{k=1}^N \sim \pi_\phi(\cdot|s')$
6: $\quad$ $Q_\theta$**- value update:**
$$y(s', r) := r + \gamma \max_{a_k'} \left[ \overline{Q}_{\theta'}(s', a_k') \right] \quad (\overline{Q} \text{ given by Eq 5})$$
$$\Delta_j(s, a) := \left( \left[ \max_{\hat{a}_k} Q_{\theta_j}(s, \hat{a}_k) - Q_{\theta_j}(s, a) \right]_+ \right)^2$$
$$\theta_j \leftarrow \operatorname*{argmin}_{\theta_j} \sum_{(s,a,s') \in \mathcal{B}} \left[ \left( Q_{\theta_j}(s,a) - y(s', r) \right)^2 + \eta \cdot \Delta_j(s,a) \right] \text{ for } j = 1,...,M$$

7: $\quad$ $\pi_\phi$ **- policy update:**
$$\phi \leftarrow \operatorname*{argmax}_\phi \sum_{(s,a) \in \mathcal{B}, \hat{a} \sim \pi_\phi(\cdot|s)} \left[ \overline{Q}_\theta(s, \hat{a}) + \lambda \cdot \log \pi_\phi(a|s) \right]$$

8: $\quad$ **Update Target Networks:**
$$\theta_j' \leftarrow \tau \theta_j + (1 - \tau) \theta_j' \ \ \forall j \in M$$
9: **end for**

---

ties introduced throughout via mini-batch estimates derived from samples taken from $\mathcal{D}$, and each optimization is performed via a few stochastic gradient method iterates.

To account for epistemic uncertainty due to the limited data, the value update in Step 6 of Algorithm 1 uses $\overline{Q}_\theta$ from (5) in place of $Q_\theta$. In CDC, we can simply utilize the same moderately conservative value of $\nu = 0.75$ used by [17], since we are not purely relying on the lower confidence bound $\overline{Q}_\theta$ to correct all overestimation. For this reason, CDC is able to achieve strong performance with a small ensemble of $M = 4$ Q-networks (used throughout this work), whereas [18] require larger ensembles of 16 Q-networks and an extremely conservative $\nu = 1$ in order to achieve good performance.

To correct extra-overestimation within each of the $M$ individual Q-networks, Algorithm 1 actually applies a separate extra-overestimation penalty $\Delta_j$ specific to each Q-network. The steps of our proposed CDC method are detailed in Algorithm 1. In blue, we highlight the only modifications CDC makes to a standard off-policy actor-critic framework that has been suitably adapted for continuous batch RL via the aforementioned techniques like EMaQ [18] and lower-confidence bounds for Q-

values [17]. Throughout, we use $\eta = 0 \ \& \ \lambda = 0$ to refer to this baseline framework (without our proposed penalties), and note that majority of modern batch RL methods like CQL [29], BCQ [17], BEAR [28], BRAC [63] are built upon similar frameworks.

Although each of our proposed regularizers can be used independently and their implementation is modular, we emphasize that they complement each other: the Q-Value regularization mitigates extra-overestimation error while the policy regularizer ensures candidate policies do not stray too far from the offline data. Ablation studies show that the best performance is only achieved through simultaneous use of both regularizers (Figure 2a, Table S1). Note that CDC is quite simple to implement: each penalty can be added to existing actor-critic RL frameworks with minimal extra code and the addition of both penalties involves no further complexity beyond the sum of the parts.

**Theorem 1** *For $\overline{Q}_\theta$ in (5), let $\mathcal{T}_{CDC} : \overline{Q}_{\theta_t} \to \overline{Q}_{\theta_{t+1}}$ denote the operator corresponding to the $\overline{Q}_\theta$-updates resulting from the $t^{th}$ iteration of Steps 6-7 of aAlgorithm 1. $\mathcal{T}_{CDC}$ is a $L_\infty$ contraction under standard conditions that suffice for the ordinary Bellman operator to be contractive [3, 6, 8, 56].*

The proof and formal list of assumptions are in Appendix D.1. Together with Banach's theorem, the contraction property established in Theorem 1 above guarantees that our CDC updates converge to a fixed point under commonly-assumed conditions that suffice for standard RL algorithms to converge [31]. Due to issues of (nonconvex) function approximation, it is difficult to guarantee this in practice or empirical optimality of the resulting estimates [38, 40]. We do note that the addition of our two novel regularizers further enhances the contractive nature and stability of the CDC updates when $\eta, \lambda > 0$ by shrinking $Q$-values and policy action-probabilities toward the corresponding values estimated for the behavior policy (i.e. values computed for observations in $\mathcal{D}$). Our CDC penalties can thus not only lead to less wildly-extrapolated batch estimates, but also faster (and more stable) convergence of the learning process (as shown in Figure 1, where *Standard actor-critic* refers to Algorithm 1 where $\eta = \lambda = 0$).

**Theorem 2** *Let $\pi_\phi \in \Pi$ be the policy learned by CDC, $\gamma$ denote discount factor, and $n$ denote the sample size of dataset $\mathcal{D}$ generated from $\pi_b$. Also let $J(\pi)$ represent the true expected return produced by deploying policy $\pi$ in the environment. Under mild assumptions listed in Appendix D, there exist constants $r^*, C_\lambda, V$ such that with high probability $\geq 1 - \delta$:*

$$J(\pi_\phi) \geq J(\pi_b) - \frac{r^*}{(1-\gamma)^2} \sqrt{C_\lambda + \sqrt{(V - \log \delta)/n}}$$

Appendix D.2 contains a proof and descriptions of the assumptions in this result. Theorem 2 assures us of the reliability of the policy $\pi_\phi$ produced by CDC, guaranteeing that with high probability $\pi_\phi$ will not have much worse outcomes than the behavior policy $\pi_b$, where the probability here depends on the size of the dataset $\mathcal{D}$ and our choice of policy regularization penalty $\lambda$ (the constant $C_\lambda$ is a decreasing function of $\lambda$). In batch settings, expecting to learn the optimal policy is futile from limited data. Even ensuring *any* improvement at all over an arbitrary $\pi_b$ is ambitious when we cannot ever test any policies in the environment, and reliability of the learned $\pi_\phi$ is thus a major concern.

**Theorem 3** *Let $\mathrm{OE}_{ag} = \mathbb{E}[\max_a Q_\theta(s, a)] - \max_a Q^*(s, a)$ be the overestimation error in actions favored by an agent ag. Here $Q_\theta$ denotes the estimate of true Q-value learned by ag, which may either use **CDC** (with $\eta > 0$) or a **baseline** version of Algorithm 1 with $\eta = 0$ (with the same value of $\lambda$). Under the assumptions listed in Appendix D.3, there co-exist constants $L_1$ and $L_2$ such that*

$$\mathrm{OE}_{\mathrm{CDC}} \leq L_1 - \eta L_2 \leq \mathrm{OE}_{\mathrm{baseline}}$$

This theorem (proved in Appendix D.3) underscores the influence of the $\eta$ parameter in terms of containing the overestimation problem in offline Q-learning. Mitigating this overestimation, which can be done using non-zero $\eta$, can ultimately lead into better returns as we show in the experimental section. In particular, CDC achieves lower overestimation by deliberately underestimating $Q$-values for non-observed state-actions (but it limits the degree of downward bias as described in Remark 2). Buckman et al. [7], Jin et al. [25] prove that some degree of pessimism is unavoidable to ensure non-catastrophic deployment of batch RL in practice, where it is unlikely there will ever be sufficient data for the agent to accurately estimate the consequences of all possible actions in all states.

**Remark 2 (Pessimism is limited in CDC)** *Extreme pessimism leads to overly conservative policies with limited returns. The degree of pessimism in CDC remains limited (capped once $\Delta_j = 0$), unlike lower-confidence bounds which can become arbitrarily pessimistic and hence limited in their return.*

# 4 Related Work

Aiming for a practical framework to improve arbitrary existing policies, much research has studied batch RL [34, 36] and the issue of overestimation [22, 23, 58]. [26, 64] consider model-based approaches for batch RL, and [1] find ensembles partly address some of the issues that arise in batch settings. To remain suitably conservative, a popular class of approaches constrain the policy updates to remain in the vicinity of $\pi_b$ via, e.g., distributional matching [17], support matching [28, 63], imposition of a behavior-based prior [52], or implicit constraints via selective policy-updates [46, 60]. Similar to imitation learning in online setting [24, 43, 47, 49], many of such methods need to explicitly estimate the behavior policy [17, 18, 28]. Although methods like [46, 60] do not have an explicit constraint on the policy update, they still can be categorized as a policy constrained-based approach as the policy update rule has been changed in a such a way that it selectively updates the policy utilizing information contained in the Q-values. Although these approaches show promising results, policy-constraint methods often work best for data collected from a high-quality (expert) behavior policy, and may struggle to significantly improve upon highly suboptimal $\pi_b$. Compared to the previous works, our CDC does not need to severely constrain candidate policies around $\pi_b$, which reduces achievable returns. Even with a strong policy constraint, the resulting policy is still affected by the learned Q-value, thus we still must correct Q-value issues. Instead of constraining policy updates, [29] advocate conservatively lower-bounding estimates of the value function. This allows for more flexibility to improve upon low-quality $\pi_b$. [39] considers a pessimistic and conservative approach to update Q-value by utilizing the marginalized state-action distribution of available data. Our proposed CDC algorithm is inspired by ideas from both the policy-constraint and value-constraint literature, demonstrating these address complementary issues of the batch RL problem and are both required in a performant solution.

# 5 Experiments

In this section, we evaluate our CDC algorithm against existing methods on 32 tasks from the D4RL benchmark [14]. We also investigate the utility of individual CDC regularizers through ablation analyses, and demonstrate the broader applicability of our extra-overestimation penalty to off-policy evaluation in addition to batch RL. Our training/evaluation setup exactly follows existing work [14, 17, 28, 29]. See Appendices A, B, and C for a complete description of our experimental pipeline.

**Setup.** We compare CDC against existing batch RL methods: BEAR [28], BRAC-V/P [63], BC [63], CQL [29], BCQ [17], RBVE[2] [20] , and SAC [21]. This covers a rich set of strong batch RL methods ranging from behavioral cloning to value-constrained-based pessimistic methods, with the exception of SAC. SAC is a popular off-policy method that empirically performs quite well in online RL, and is included to investigate how online RL methods fare when applied to the batch setting. Note that CDC was simply run on every task using the same network and the original rewards/actions provided in the task, without any manual task-specific reward-normalization/action-smoothing. Moreover, all these baseline methods also utilize an ensemble of Q networks as in (5).

**Results.** Figure 2b and Table 1 illustrate that CDC performs better than the majority of the other batch RL methods on the D4RL tasks. Across all 32 tasks, CDC obtains normalized return of 1397, whereas the next-best method (CQL) achieves 1245. In head-to-head comparisons, CDC generates statistically significantly greater overall returns (Table 1). Unsurprisingly, behavioral-cloning (BC) works well on tasks with data generated by an expert $\pi_b$, while the online RL method, SAC, fares poorly in many tasks. CDC remains reasonably competitive across all tasks, regardless of the environment or the quality of $\pi_b$ (i.e. random vs. expert).

Next we perform a comprehensive set of ablation studies to gauge the contribution of our proposed penalties in CDC. Here we run additional variants of Algorithm 1 without our penalties (i.e. $\eta = \lambda = 0$) , with only our extra-overestimation penalty ($\lambda = 0$), and with only our exploration penalty ($\eta = 0$). Figure 2a and Tables S1 show that both penalties are critical for the strong performance of CDC, with the extra-overestimation penalty $\Delta$ being of greater importance than exploration (see also Figure 2a). Note that *all* our ablation variants still employ the lower confidence bound from (5), which alone clearly does not suffice to correct extra-overestimation.

---

[2]Comparison between CDC and the concurrently-proposed RBVE method [20] is relegated to Section A.1.

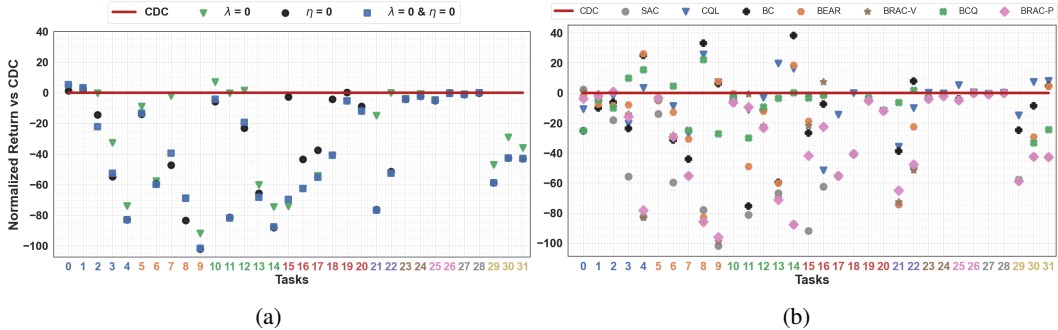

(a)                      (b)

Figure 2: **Difference in (normalized) return achieved by various algorithms vs CDC in 32 D4RL tasks**. X-axis colors indicate environments (see Table 1), and points below the line ( ▬▬ ) indicate worse performance than CDC. **Figure 2a** shows that fixing $\eta$ or $\lambda$ to zero (i.e. omitting our penalties) produces far worse returns than CDC (see also Table S1). This ablation study proves that major performance gains for CDC stem from our novel pair of regularizers, as the *only* difference between CDC and these ablated variants is either $\eta$ or $\lambda$ or both are set to zero in Algorithm 1 (all other details are exactly the same). **Figure 2b** compares CDC against existing batch RL algorithms, where CDC overall compares favorably to each other method in head-to-head comparisons (see also Table 1). Note these figures can be compared to each other as well.

## 5.1 Offline Policy Evaluation

The true practical applicability of batch RL remains however hampered without the ability to do proper algorithm/hyperparameter selection. Table 1 shows that no algorithm universally dominates all others across all environments or behavior-policies. In practice, it is difficult to know which technique will perform best, unless one can do proper offline policy evaluation (OPE) of different candidate policies before their actual online deployment [15, 45].

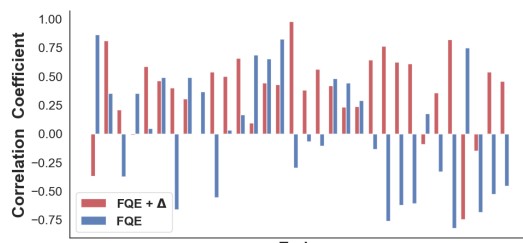

Figure 3: **How well OPE estimates correlate with actual return** achieved by 20 different policies for each D4RL task. Due to unmitigated overestimation, FQE estimates correlate *negatively* with true returns in 15 of 32 tasks (using $\Delta$ in FQE reduces this to 4).

OPE aims to estimate the performance of a given policy under the same setting considered here, with offline data collected by an unknown behavior policy [35, 45]. Beyond algorithm/hyperparameter comparison, OPE is often employed for critical policy-making decisions where environment interaction is no longer an option, e.g., sensitive healthcare applications [19]. One practical OPE method for data of the form in $\mathcal{D}$ is Fitted Q Evaluation (FQE) [35]. To score a given policy $\pi$, FQE iterates temporal difference updates of the form (3) using the standard Bellman operator from (1) in place of EMaQ. After learning an estimate $\hat{Q}^\pi$, FQE simply estimates the return of $\pi$ via the expectation of $\hat{Q}^\pi(s,a)$ over the initial state distribution and actions sampled from $\pi$.

However, like batch RL, OPE also relies on limited data and thus can still suffer from severe Q-value estimation errors. To contain the overestimation bias, we can regularize the FQE temporal difference updates with our $\Delta$ penalty, in a similar manner to (7). Figure 3 compares the performance of $\Delta$-penalization of FQE (with $\eta = 1$ throughout) against the standard unregularized FQE. Here we use both OPE methods to score 20 different policies (learned via different settings) and gauge OPE-quality via the Pearson correlation between OPE estimated returns and the actual return (over our 20 policies). We observe higher correlation for FQE + $\Delta$ (0.37 on average) over FQE (0.01 on average) in the majority of tasks, demonstrating the usefulness of our regularizers. The usefulness of our regularizers thus extend beyond batch RL and carry to the off-policy evaluation setting.

---

[1]The results for CQL are taken from the official author-provided codes [`https://github.com/ aviralkumar2907/CQL`] of [29]. The published CQL codes are used to produce results for all but Adroit and FrankaKitchen where the codes are not available. For these latter domains, we simply use the CQL results reported in the paper of [29].

| Index | Task Name | SAC | BC | BRAC-P | BRAC-V | BEAR | BCQ | CQL[1] | $\lambda = 0$ & $\eta = 0$ | CDC |
|---|---|---|---|---|---|---|---|---|---|---|
| 0 | halfcheetah-random | 29.6 | 2.1 | 23.5 | 28.1 | 25.5 | 2.25 | 16.71 | **32.8** | 27.36 |
| 1 | halfcheetah-medium | 40.97 | 36.1 | 44.0 | 45.5 | 38.6 | 41.48 | 38.97 | **49.51** | 46.05 |
| 2 | halfcheetah-medium-replay | 26.47 | 38.4 | 45.6 | **45.9** | 36.2 | 34.79 | 42.77 | 22.72 | 44.74 |
| 3 | halfcheetah-medium-expert | 3.78 | 35.8 | 43.8 | 45.3 | 51.7 | **69.64** | 39.18 | 7.12 | 59.64 |
| 4 | halfcheetah-expert | $-0.41$ | 107.0 | 3.8 | $-1.1$ | **108.2** | 97.44 | 85.49 | $-0.95$ | 82.05 |
| 5 | hopper-random | 0.8 | 9.8 | 11.1 | 12.0 | 9.5 | 10.6 | 10.37 | 1.58 | **14.76** |
| 6 | hopper-medium | 0.81 | 29.0 | 31.2 | 32.3 | 47.6 | **65.07** | 51.79 | 0.58 | 60.39 |
| 7 | hopper-medium-replay | 0.59 | 11.8 | 0.7 | 0.8 | 25.3 | 31.05 | 28.67 | 16.4 | **55.89** |
| 8 | hopper-medium-expert | 8.96 | **119.9** | 1.1 | 0.8 | 4.0 | 109.1 | 112.46 | 18.07 | 86.9 |
| 9 | hopper-expert | 0.8 | 109.0 | 6.6 | 3.7 | **110.3** | 75.52 | 109.97 | 1.27 | 102.75 |
| 10 | walker2d-random | 1.3 | 1.6 | 0.8 | 0.5 | 6.7 | 4.31 | 2.77 | 2.96 | **7.22** |
| 11 | walker2d-medium | 0.81 | 6.6 | 72.7 | 81.3 | 33.2 | 52.03 | 71.03 | 0.33 | **82.13** |
| 12 | walker2d-medium-replay | 0.04 | 11.3 | $-0.3$ | 0.9 | 10.8 | 13.67 | 19.95 | 3.81 | **22.96** |
| 13 | walker2d-medium-expert | 4.09 | 11.3 | $-0.3$ | 0.9 | 10.8 | 67.26 | **90.55** | 2.65 | 70.91 |
| 14 | walker2d-expert | 0.05 | **125.7** | $-0.2$ | $-0.0$ | 106.1 | 87.59 | 103.6 | $-0.1$ | 87.54 |
| 15 | antmaze-umaze | 0.0 | 65 | 50 | 70 | 73 | 88.52 | 90.12 | 22.22 | **91.85** |
| 16 | antmaze-umaze-diverse | 0.0 | 55 | 40 | **70** | 61 | 61.11 | 11.11 | 0.0 | 62.59 |
| 17 | antmaze-medium-play | 0.0 | 0 | 0 | 0 | 0 | 0.0 | 40.74 | 0.0 | **55.19** |
| 18 | antmaze-medium-diverse | 0.0 | 0 | 0 | 0 | 0 | 0.0 | 40.74 | 0.0 | **40.74** |
| 19 | antmaze-large-play | 0.0 | 0 | 0 | 0 | 0 | 1.85 | 0.0 | 0.0 | **5.19** |
| 20 | antmaze-large-diverse | 0.0 | 0 | 0 | 0 | 0 | 0.0 | 0.0 | 0.0 | **11.85** |
| 21 | pen-human | $-1.15$ | 34.4 | 8.1 | 0.6 | $-1.0$ | 66.88 | 37.5 | $-3.43$ | **73.19** |
| 22 | pen-cloned | $-0.64$ | **56.9** | 1.6 | $-2.5$ | 26.5 | 50.86 | 39.2 | $-3.4$ | 49.18 |
| 23 | hammer-human | 0.26 | 1.5 | 0.3 | 0.2 | 0.3 | 0.91 | **4.4** | 0.26 | 4.34 |
| 24 | hammer-cloned | 0.27 | 0.8 | 0.3 | 0.3 | 0.3 | 0.38 | 2.1 | 0.26 | **2.37** |
| 25 | door-human | $-0.34$ | 0.5 | $-0.3$ | $-0.3$ | $-0.3$ | $-0.05$ | **9.9** | $-0.16$ | 4.62 |
| 26 | door-cloned | $-0.34$ | $-0.1$ | $-0.1$ | $-0.1$ | $-0.1$ | 0.01 | **0.4** | $-0.36$ | 0.01 |
| 27 | relocate-human | $-0.31$ | 0 | $-0.3$ | $-0.3$ | $-0.3$ | $-0.04$ | 0.2 | $-0.31$ | **0.73** |
| 28 | relocate-cloned | $-0.11$ | **$-0.1$** | $-0.3$ | $-0.3$ | $-0.2$ | $-0.28$ | $-0.1$ | $-0.15$ | $-0.24$ |
| 29 | kitchen-complete | 0.0 | 33.8 | 0 | 0 | 0 | 0.83 | 43.8 | 0.0 | **58.7** |
| 30 | kitchen-partial | 0.0 | 33.8 | 0 | 0 | 13.1 | 9.26 | **49.8** | 0.0 | 42.5 |
| 31 | kitchen-mixed | 0.0 | 47.5 | 0 | 0 | 47.2 | 18.43 | **51** | 0.0 | 42.87 |
| | Total Score | 116.28 | 984.4 | 383.4 | 434.5 | 844.0 | 1060.46 | 1245.2 | 173.67 | **1396.99** |
| | **p-value vs. CDC** | 7.0e-07 | 1.6e-03 | 5.3e-07 | 3.9e-06 | 1.1e-04 | 6.1e-04 | 3.6e-02 | 1.8e-06 | - |

Table 1: **Return achieved in deployment of policies learned via different batch RL methods.** The return in each environment here is normalized using (12) as originally advocated by [14]. For each method: we perform a head-to-head comparison against CDC across the D4RL tasks, reporting the $p$-value of a (one-sided) Wilcoxon signed rank test [61] that compares this method's return against that of CDC (over the 32 tasks). Here $\lambda = 0$ & $\eta = 0$ is variant of Algorithm 1 without our penalties where it proves that major performance gains for CDC stem from our novel pair of regularizers.

## 6 Discussion

Here we propose a simple and effective algorithm for batch RL by introducing a simple pair of regularizers that abate the challenge of learning how to act from limited data. The first constrains the value update to mitigate extra-overestimation error, while the latter constrains the policy update to ensure candidate policies do not stray too far from the offline data. Unlike previous work, this paper highlights the utility of simultaneous policy and value regularization in batch RL. One can envision other combinations of alternative policy and value regularizers that may perform even better than the particular policy/value penalties used in CDC. That said, our CDC penalties are particularly simple to incorporate into arbitrary actor-critic RL frameworks, and operate synergistically as illustrated in the ablation studies. Comprehensive experiments on standard offline continuous-control benchmarks suggest that CDC compares favorably with state-of-the-art methods for batch RL, and our proposed penalties are also useful to improve offline policy evaluation. The broader impact of this work will hopefully be to improve batch RL performance in offline applications, but we caution that unobserved confounding remains another key challenge in real-world data that was not addressed in this work.

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
