# Appendix: Continuous Doubly Constrained Batch Reinforcement Learning

## A  Experiment Details

**Evaluation Procedure.**   We measure performance in each task using the rewards collected by the learned policy when actually deployed in the environment. To report more stable results, we follow [29] and average returns achieved by each of the policies arising during the last 10k gradient steps of batch RL (done for all batch RL methods). To further improve stability, we also re-run all batch RL methods with 3 different random seeds and take another average across the resulting performance. As suggested by [14], we report returns for each task that have been normalized as follows:

$$\text{score} = 100 * \frac{\text{score} - \text{random score}}{\text{expert score} - \text{random score}} \tag{12}$$

where *random score* and *expert score* are provided for each task by [14] in the D4RL paper GitHub repository[3]. The same procedure is also used in previous works [14, 29] to report results and compare various batch RL methods.

**Baselines.**   We compare CDC against standard baselines and state-of-the-art batch RL methods: BEAR [28], BRAC-V and BRAC-P [63], BC [63], CQL [29], BCQ [17], and SAC [21]. We obtained the numbers for BEAR, BC, BRAC-V, and BRAC-P from published numbers by  [29]. However, numbers for BCQ and SAC are from our runs for all tasks. Also, we run published CQL codes[4] with their hyperparameters to produce results for all but Adroit and FrankaKitchen where the codes are not available. For these latter domains, we simply use the CQL results reported in the paper of [29]. In head-to-head comparisons against each of these other batch RL methods, CDC generates greater overall returns (see Table 1). To verify these results are statistically significant, we report the $p$-value of a (one-sided) Wilcoxon signed rank test [61] comparing the returns of another method vs the returns of CDC across all 32 datsets (see **p-value vs. CDC** row in Table 1).

To provide a better picture of our method, we also include the learning curves in Figure S1 for our algorithm vs BCQ for each environment considered in our benchmark. These plots show that, in the vast majority of environments, CDC exhibits consistently better performance across different seeds/iterations.

**Ablation studies.**   We conduct a series of ablation studies to comprehensively analyze the different components of CDC. We use all 32 D4RL datasets for this purpose. Table S1 and Table S2 show that both penalties introduced in our paper are critical for the strong performance of CDC, with the extra-overestimation penalty $\Delta$ being of greater importance than the exploration-penalty $\log \pi$. Moreover, Figure S2 shows how estimated Q values evolve over training for each of the above ablation variants. Here it is again evident that both penalties may be required to successfully prevent extra-overestimation and subsequent explosion of Q-estimates, with $\Delta$ being the more effective of the two for mitigating extra-overestimation.

### A.1  Comparing CDC with RBVE (Gulcehre et al. [20])

Concurrent to our work, Gulcehre et al. [20] propose a value regularization term for batch RL that is similar to the Q-value regularizer used by CDC. Unlike our work, [20] only considers discrete actions under a DQN [41] framework rather than the actor-critic RL framework employed in CDC. [20] also does not consider explicit policy regularization, which forms a critical component of CDC to supplement its value regularization. That said, [20] do also acknowledge the importance of ensuring the learned policy does not stray too far from the behavior policy, but their proposal to ensure this involves restricting the learner to apply only a single step of policy-iteration to the estimated value function. However in continuous action spaces with a policy-network, even a single policy-iteration step can lead to large deviations from the behavior policy without explicit policy regularization as imposed by CDC. Finally, we note that the RBVE methodology of [20] requires a dataset that

---

[3]`https://github.com/rail-berkeley/d4rl/blob/master/d4rl/infos.py`
[4]`https://github.com/aviralkumar2907/CQL`.

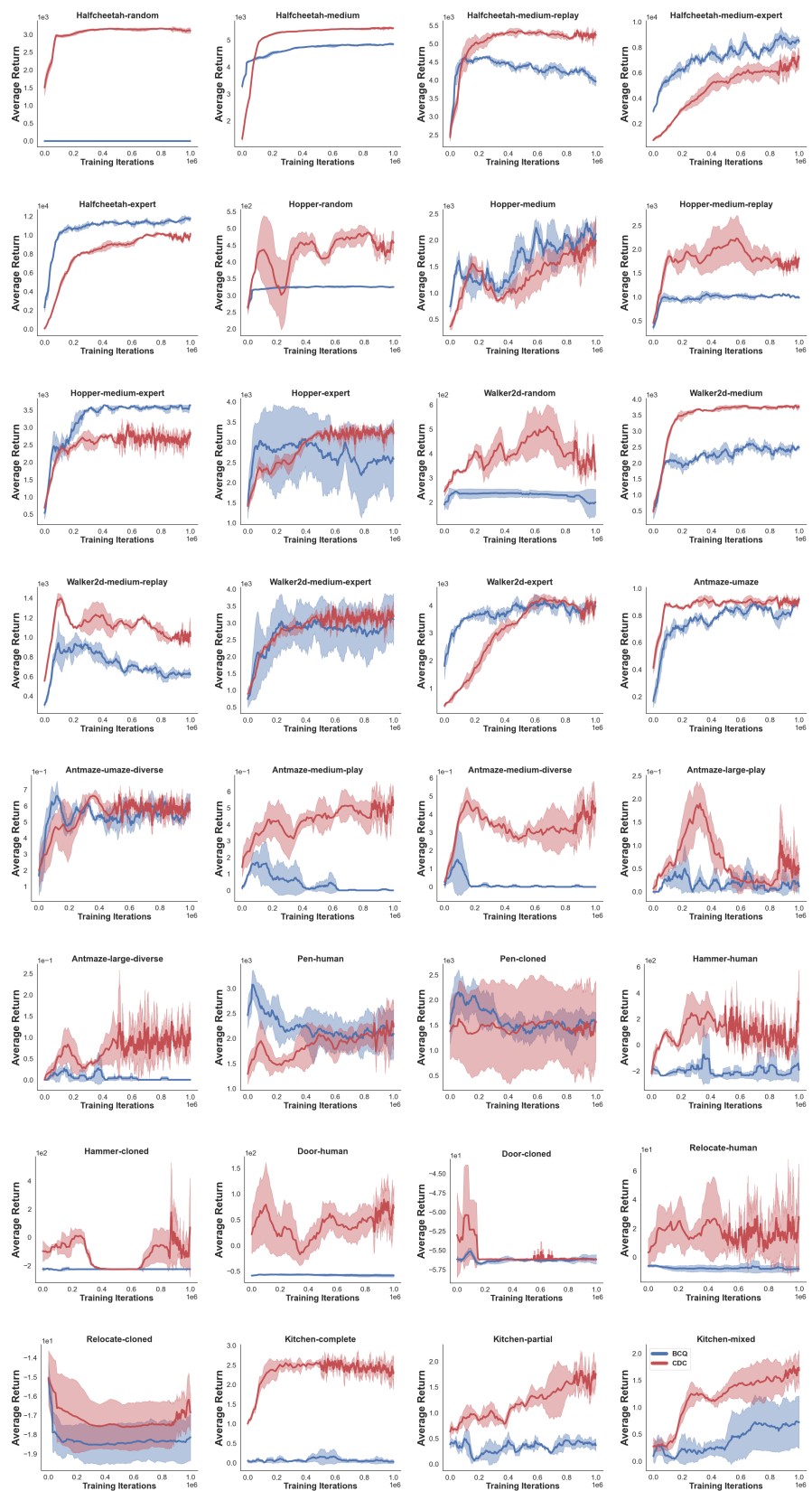

Figure S1: Learning curves of CDC (red) and BCQ (blue) on all 32 D4RL environments. Curves are averaged over 3 seeds, with the shaded area showing the standard deviation across seeds.

| Index | Task Name | $\lambda = 0$ & $\eta = 0$ | $\eta = 0$ | $\lambda = 0$ | CDC |
|-------|-----------|----------------------------|------------|---------------|-----|
| 0 | halfcheetah-random | **32.8** | 28.78 | 30.66 | 27.36 |
| 1 | halfcheetah-medium | **49.51** | 48.25 | 47.61 | 46.05 |
| 2 | halfcheetah-medium-replay | 22.72 | 30.47 | 44.62 | **44.74** |
| 3 | halfcheetah-medium-expert | 7.12 | 4.98 | 26.99 | **59.64** |
| 4 | halfcheetah-expert | $-0.95$ | $-0.96$ | 8.21 | **82.05** |
| 5 | hopper-random | 1.58 | 0.84 | 5.97 | **14.76** |
| 6 | hopper-medium | 0.58 | 1.05 | 2.71 | **60.39** |
| 7 | hopper-medium-replay | 16.4 | 8.8 | 54.01 | **55.89** |
| 8 | hopper-medium-expert | 18.07 | 3.64 | 18.22 | **86.9** |
| 9 | hopper-expert | 1.27 | 0.8 | 10.89 | **102.75** |
| 10 | walker2d-random | 2.96 | 1.55 | **14.37** | 7.22 |
| 11 | walker2d-medium | 0.33 | 0.85 | 81.93 | **82.13** |
| 12 | walker2d-medium-replay | 3.81 | $-0.14$ | **24.48** | 22.96 |
| 13 | walker2d-medium-expert | 2.65 | 5.22 | 10.94 | **70.91** |
| 14 | walker2d-expert | $-0.1$ | $-0.42$ | 13.03 | **87.54** |
| 15 | antmaze-umaze | 22.22 | 89.26 | 17.78 | **91.85** |
| 16 | antmaze-umaze-diverse | 0.0 | 19.26 | 0.0 | **62.59** |
| 17 | antmaze-medium-play | 0.0 | 17.78 | 1.11 | **55.19** |
| 18 | antmaze-medium-diverse | 0.0 | 36.67 | 0.0 | **40.74** |
| 19 | antmaze-large-play | 0.0 | **5.56** | 0.0 | 5.19 |
| 20 | antmaze-large-diverse | 0.0 | 2.96 | 0.0 | **11.85** |
| 21 | pen-human | $-3.43$ | $-3.07$ | 58.33 | **73.19** |
| 22 | pen-cloned | $-3.4$ | $-2.29$ | **49.31** | 49.18 |
| 23 | hammer-human | 0.26 | 0.26 | 0.66 | **4.34** |
| 24 | hammer-cloned | 0.26 | 0.28 | 1.79 | **2.37** |
| 25 | door-human | $-0.16$ | $-0.34$ | 0.01 | **4.62** |
| 26 | door-cloned | $-0.36$ | $-0.13$ | **0.14** | 0.01 |
| 27 | relocate-human | $-0.31$ | $-0.31$ | 0.0 | **0.73** |
| 28 | relocate-cloned | **$-0.15$** | $-0.34$ | $-0.25$ | $-0.24$ |
| 29 | kitchen-complete | 0.0 | 0.0 | 11.76 | **58.7** |
| 30 | kitchen-partial | 0.0 | 0.0 | 13.52 | **42.5** |
| 31 | kitchen-mixed | 0.0 | 0.0 | 7.04 | **42.87** |
| | Total Score | 173.67 | 299.25 | 555.84 | **1396.99** |

Table S1: **Ablation study of components used in CDC.** Listed is the return in each environment (normalized using (12) as in [14]) achieved by ablated variants of our algorithm. Fixing $\eta$ or $\lambda$ to zero (i.e. omitting our penalties) produces far worse returns than CDC, demonstrating the utility of both of our proposed penalties. Note that the **only** difference between CDC and these variants (i.e. $\lambda = 0$ & $\eta = 0$, $\eta = 0$, $\lambda = 0$) in these experiments is either $\eta$ or $\lambda$ or both are set to zero in Algorithm 1 and all other details are **exactly** the same.

contains observations $(s, a, r, s', a')$, i.e. more complete subtrajectories of episodes, whereas CDC merely requires a dataset that contains observations of the form $(s, a, r, s')$. The former setting is less widely applicable, but is somewhat easier due to the availability of the subsequent action $a'$ for temporal-difference learning.

In this section, we apply the RBVE method of [20] on the D4RL benchmark [14], after first minorly adapting it to our setting. Key differences in our setting are: we have continuous actions, and $a'$ is not contained in the dataset. In our adaptation of RBVE, we approximate the $\max_a$ required by [20] (but which is difficult for continuous actions) by sampling many actions and taking the empirical maximum. Our adaptation accounts for the fact that $a'$ is not present in the dataset by first estimating the behavior policy $\pi_b$ via behavior-cloning (i.e. via maximum likelihood training of our same policy network) and then drawing $a' \sim \pi_b(.|s')$ for use in RBVE. Furthermore, we considered two different variants of RBVE in our experiments. In the first variant (called RBVE-A), the soft filtering weights of [20], $\omega(s, a)$, are implemented according to Eq 6 in their paper. Although closely following the recommendations of [20], RBVE-A did not perform well in our D4RL environments, and thus we considered a second variant (called RBVE-C), where we treat $\omega$ as a hyperparameter and we use a fixed value per environment. Table S3 illustrates that CDC outperforms both variants of RBVE.

| Task Name | $\lambda = 0$ & $\eta = 0$ | $\eta = 0$ | $\lambda = 0$ | CDC |
|---|---|---|---|---|
| halfcheetah-random | **3791.65** | 3293.31 | 3526.06 | 3117.23 |
| halfcheetah-medium | **5865.97** | 5710.67 | 5631.1 | 5437.01 |
| halfcheetah-medium-replay | 2540.07 | 3503.24 | 5259.18 | **5274.51** |
| halfcheetah-medium-expert | 604.05 | 337.88 | 3070.63 | **7124.4** |
| halfcheetah-expert | $-398.72$ | $-399.66$ | 739.54 | **9906.71** |
| hopper-random | 31.08 | 7.19 | 174.13 | **459.99** |
| hopper-medium | $-1.48$ | 13.77 | 68.07 | **1945.29** |
| hopper-medium-replay | 513.53 | 266.05 | 1737.61 | **1798.66** |
| hopper-medium-expert | 567.85 | 98.11 | 572.74 | **2808.05** |
| hopper-expert | 20.98 | 5.92 | 334.25 | **3323.93** |
| walker2d-random | 137.71 | 72.86 | **661.34** | 333.2 |
| walker2d-medium | 16.82 | 40.53 | 3762.54 | **3771.93** |
| walker2d-medium-replay | 176.51 | $-4.86$ | **1125.45** | 1055.62 |
| walker2d-medium-expert | 123.48 | 241.43 | 504.05 | **3257.06** |
| walker2d-expert | $-3.01$ | $-17.79$ | 599.6 | **4020.49** |
| antmaze-umaze | 0.22 | 0.89 | 0.18 | **0.92** |
| antmaze-umaze-diverse | 0.0 | 0.19 | 0.0 | **0.63** |
| antmaze-medium-play | 0.0 | 0.18 | 0.01 | **0.55** |
| antmaze-medium-diverse | 0.0 | 0.37 | 0.0 | **0.41** |
| antmaze-large-play | 0.0 | **0.06** | 0.0 | 0.05 |
| antmaze-large-diverse | 0.0 | 0.03 | 0.0 | **0.12** |
| pen-human | $-5.87$ | 4.84 | 1834.73 | **2277.75** |
| pen-cloned | $-5.04$ | 28.09 | **1565.86** | 1562.21 |
| hammer-human | $-241.31$ | $-241.06$ | $-188.27$ | **292.21** |
| hammer-cloned | $-241.4$ | $-238.34$ | $-41.52$ | **34.45** |
| door-human | $-61.11$ | $-66.61$ | $-56.14$ | **79.05** |
| door-cloned | $-67.09$ | $-60.23$ | **$-52.37$** | $-56.15$ |
| relocate-human | $-19.51$ | $-19.72$ | $-6.37$ | **24.36** |
| relocate-cloned | **$-12.77$** | $-20.95$ | $-17.05$ | $-16.63$ |
| kitchen-complete | 0.0 | 0.0 | 0.47 | **2.35** |
| kitchen-partial | 0.0 | 0.0 | 0.54 | **1.7** |
| kitchen-mixed | 0.0 | 0.0 | 0.28 | **1.71** |
| Total Score | 13332.61 | 12556.37 | 30806.65 | **57839.77** |

Table S2: **Ablation study of components used in CDC.** Same as Table S1 but the returns here are not normalized, and we instead report raw returns achieved in each task.

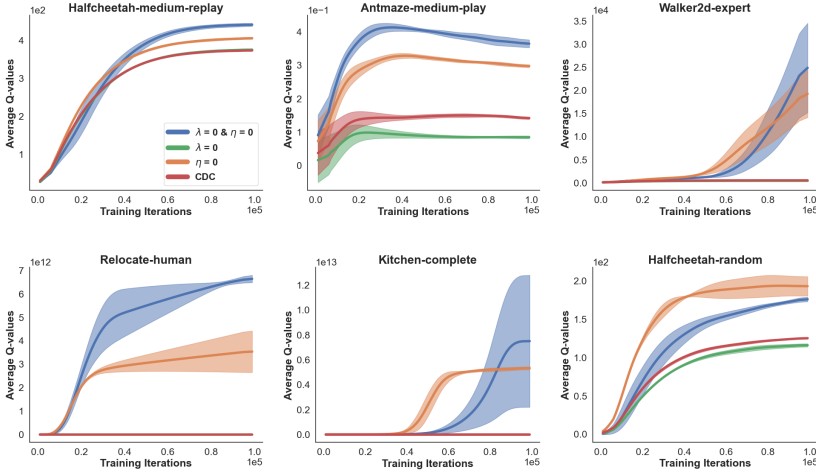

Figure S2: **Effect of our penalties on Q-values.** These figures show evaluation of averaged Q values across 4 Q during training time for 6 different tasks. This result shows that CDC's Q-estimate is well controlled especially compared with $\eta = 0$.

| Index | Task Name | RBVE-C | RBVE-A | CDC |
|---|---|---|---|---|
| 0 | halfcheetah-random | 18.89 | -0.01 | **27.36** |
| 1 | halfcheetah-medium | 43.98 | 24.27 | **46.05** |
| 2 | halfcheetah-medium-replay | 37.24 | 7.55 | **44.74** |
| 3 | halfcheetah-medium-expert | 32.11 | 8.12 | **59.64** |
| 4 | halfcheetah-expert | 36.57 | 3.14 | **82.05** |
| 5 | hopper-random | 11.46 | 4.69 | **14.76** |
| 6 | hopper-medium | 17.16 | 1.23 | **60.39** |
| 7 | hopper-medium-replay | 28.15 | 3.31 | **55.89** |
| 8 | hopper-medium-expert | **88.72** | 3.62 | 86.9 |
| 9 | hopper-expert | 94.32 | 1.2 | **102.75** |
| 10 | walker2d-random | 0.36 | 2.4 | **7.22** |
| 11 | walker2d-medium | 80.19 | 2.81 | **82.13** |
| 12 | walker2d-medium-replay | 6.7 | 2.41 | **22.96** |
| 13 | walker2d-medium-expert | **77.79** | 1.83 | 70.91 |
| 14 | walker2d-expert | 60.3 | 0.69 | **87.54** |
| 15 | antmaze-umaze | 0.0 | 0.0 | **91.85** |
| 16 | antmaze-umaze-diverse | 2.96 | 0.0 | **62.59** |
| 17 | antmaze-medium-play | 0.0 | 0.0 | **55.19** |
| 18 | antmaze-medium-diverse | 0.0 | 0.0 | **40.74** |
| 19 | antmaze-large-play | 0.0 | 0.0 | **5.19** |
| 20 | antmaze-large-diverse | 0.0 | 0.0 | **11.85** |
| 21 | pen-human | 24.04 | 34.93 | **73.19** |
| 22 | pen-cloned | 42.86 | -0.39 | **49.18** |
| 23 | hammer-human | 0.59 | 0.0 | **4.34** |
| 24 | hammer-cloned | 0.34 | 0.17 | **2.37** |
| 25 | door-human | **9.15** | -0.0 | 4.62 |
| 26 | door-cloned | **0.04** | 0.03 | 0.01 |
| 27 | relocate-human | 0.29 | 0.01 | **0.73** |
| 28 | relocate-cloned | -0.23 | **0.01** | -0.24 |
| 29 | kitchen-complete | 18.61 | 0.83 | **58.7** |
| 30 | kitchen-partial | 8.7 | 0.83 | **42.5** |
| 31 | kitchen-mixed | 5.93 | 1.11 | **42.87** |
| | Total Score | 747.21 | 104.79 | **1396.99** |
| | **p-value vs. CDC** | 6.6e-06 | 5.3e-07 | - |

Table S3: **Return achieved in deployment of policies learned via CDC and RBVE [20].** RBVE-C and RBVE-A are two variants of [20] detailed in Section A.1, and we use the exact same setup for CDC as before. The return in each environment here is normalized using (12) as originally advocated by [14]. For each method: we perform a head-to-head comparison against CDC across the D4RL tasks, reporting the $p$-value of a (one-sided) Wilcoxon signed rank test [61] that compares this method's return against that of CDC (over the 32 tasks).

## B  Details of our Methods

**Implementation Details.**    Table S5 and Table S4 show hyper-parameters, computing infrastructure, and libraries used for the experiments in this paper for all 32 continuous-control tasks. Like most other batch-RL baselines in our comparisons and following Sec 8 in [1], we did a minimal random search to tune our hyperparams $\eta, \lambda$. Note that CDC was simply run on every task using the same network architecture and the original rewards/actions provided in the task, without any task-specific reward-normalization/action-smoothing required by some of the other batch RL methods.

**Using CDC Policy During Deployment.**    Algorithm 1 in the main text only describes the training process used in CDC, Algorithm 3 here details how a batch RL is deployed the resulting learned policy/values to select actions in the actual environment. After the batch RL training is complete, Algorithm 3 is used to select actions at evaluation (test) time, as also done by [17, 18, 28, 29, 63]. All CDC returns mentioned throughout the paper (and other baseline methods returns i.e. BCQ, CQL, BEAR, BRAC-V/P, etc.) were produced by selecting actions in this manner, which facilitates fair comparison against the existing literature.

**Algorithm 2** FQE + $\Delta$

1: **Input**: policy $\pi$ to evaluate
2: Initialize Q networks: $\{Q_{\theta_j}\}_{j=1}^M$
3: Initialize Target Q-networks: $\{Q_{\theta'_j} : \theta'_j \leftarrow \theta_j\}_{j=1}^M$
4: **for** $t$ in $\{1, \ldots, T\}$ **do**
5:     Sample mini-batch $\mathcal{B} = \{(s, a, r, s')\} \sim \mathcal{D}$
6:     For each $s, s' \in \mathcal{B}$: sample $N$ actions $\{\hat{a}_k\}_{k=1}^N \sim \pi(\cdot|s)$ and $\{a'_k\}_{k=1}^N \sim \pi(\cdot|s')$
7:     $Q_\theta$**- value update**:

$$y(s', r) := r + \frac{\gamma}{N} \sum_{a'_k}^N \left[ \overline{Q}_{\theta'}(s', a'_k) \right] \text{ where } \overline{Q} \text{ given by Eq 5}$$

$$\Delta_j(s, a) := \left( \left[ \max_{\hat{a}_k} Q_{\theta_j}(s, \hat{a}_k) - Q_{\theta_j}(s, a) \right]_+ \right)^2$$

$$\theta_j \leftarrow \operatorname*{argmin}_{\theta_j} \sum_{(s,a,s') \in \mathcal{B}} \left[ \left( Q_{\theta_j}(s, a) - y(s', r) \right)^2 + \eta \cdot \Delta_j(s, a) \right] \text{ for } j = 1, ..., M$$

8:     **Update Target Networks:**
    $\theta'_j \leftarrow \tau \theta_j + (1 - \tau)\theta'_j \ \ \forall j \in M$
9: **end for**

---

**Algorithm 3** Bacth RL Action Selection at Evaluation Time (used for CDC as well as other baseline methods)

1: **Input**: state $s \in S$, trained policy network $\pi_\phi$ and Q networks: $\{Q_{\theta_j}\}_{j=1}^M$.
2: Sample $N$ actions $\{a_k\}_{k=1}^N \sim \pi_\phi(\cdot|s)$
3: Identify optimal action:

$$a \leftarrow \arg\max_{a_k} \left[ \overline{Q}_\theta(s, a_k) \right]$$

Here $\overline{Q}$ given by Eq 5 (Note this similar to BCQ, BRAC-V/P, BEAR, EMaQ).
4: **Return** $a$

## B.1 Fitted Q Evaluation Details

For off-policy evaluation, Algorithm 2 describes the steps of fitted Q-evaluation (FQE) [35], when additionally leveraging our extra-overestimation penalty $\Delta$. The goal of FQE is to estimate the values for a given policy, i.e. $\hat{Q}^\pi$, with offline data collected by an unknown behavior policy. After learning an estimate $\hat{Q}^\pi$, the resulting Q-values will be used to score a policy $\pi$ via the expectation of $\hat{Q}^\pi$ over the initial state distribution and actions proposed by this policy, i.e. the estimated expected return, which is given by $\hat{v}(\pi) = \mathbb{E}_{s \sim \mathcal{D}} \mathbb{E}_{a \sim \pi(\cdot|s)}[\hat{Q}^\pi(s, a)]$ [35]. Applying Q-learning to limited data, FQE is also prone to suffer from wild extrapolation, which we attempt to mitigate by introducing our $\Delta$ penalty (highlighted blue terms in Algorithm 2 are our modifications to FQE).

In Figure 3 of Section 5.1 in the main text, we compare the performance of $\Delta$-penalization of FQE (with $\eta = 1$ throughout in Algorithm 2) against the standard unregularized FQE ($\eta = 0$ in Algorithm 2). Here we use both methods to score 20 different policies, learned under CDC with different random hyperparameter settings. When scoring each CDC-policy, $a \sim \pi$ in the definition of $\hat{v}$ is obtained using Algorithm 3 for each $s$, as the operations of Algorithm 3 entail the actual policy considered for deployment.

Here we assess the quality of FQE policy evaluation via the Pearson correlation between estimated returns, i.e., $\hat{v}(\pi)$, and the actual return (over our 20 policies under consideration). The higher correlations observed for FQE + $\Delta$ (0.37 on average across our 32 tasks) over FQE (0.01 on average) in the majority of tasks demonstrates how the inclusion of our $\Delta$ penalty can lead to more reliable off policy evaluation estimates. Our strategies for mitigating overestimation/extrapolation are thus not only useful for batch RL but also related tasks like off-policy evaluation.

## C  D4RL Benchmark

D4RL is a large-scale benchmark for evaluating batch RL algorithms [14]. It contains many diverse tasks with different levels of complexity in which miscellaneous behavior policies (ranging from random actions to expert demonstrations) have been used to collect data. For each task, batch RL agents are trained on a large offline dataset $\mathcal{D}$ (without environment interaction), and these agents are scored based on how much return they produce when subsequently deployed into the live environment. Since the benchmark contains multiple tasks from a single environment (with different $\pi_b$), we can observe how well batch RL methods are able to learn from behavior policies of different quality.

| Computing Infrastructure | |
|---|---|
| Machine Type | AWS EC2 - p2.16xlarge |
| CUDA Version | 10.2 |
| NVIDIA-Driver | 440.33.01 |
| GPU Family | Tesla K80 |
| CPU Family | Intel Xeon 2.30GHz |
| **Library Version** | |
| Pytorch | 1.6.0 |
| Gym | 0.17.2 |
| Python | 3.7.7 |
| Numpy | 1.19.1 |

Table S4: Computing infrastructure and software libraries used in this paper.

| Hyper-parameters | value |
|---|---|
| Random Seeds | $\{0, 1, 2\}$ |
| Overestimation bias coef ($\nu$) | 0.75 |
| Batch Size | 256 |
| Number of Updates | 1e+6 |
| Number of $Q$ Functions | 4 |
| Number of hidden layers (Q) | 4 layers |
| Number of hidden layers ($\pi$) | 4 layers |
| Number of hidden units per layer | 256 |
| Number of sampled actions ($N$) | 15 |
| Nonlinearity | *ReLU* |
| Discount factor ($\gamma$) | 0.99 |
| Target network ($\theta'$) update rate ($\tau$) | 0.005 |
| Actor learning rate | 3e-4 |
| Critic learning rate | 7e-4 |
| Optimizer | Adam |
| Policy constraint coef ($\lambda$) | $\{0.1, 0.5, 1, 2\}$ |
| Extra-overestimation coef ($\eta$) | $\{.1, .2, .5, .6, .8, 1, 5, 20, 25, 50, 100, 200\}$ |
| Number of episodes to evaluate | 10 |

Table S5: Hyper-parameters used for CDC for all 32 continuous-control tasks in the D4RL benchmark. All results reported in our paper are averages over repeated runs initialized with *each* of the random seeds listed above and run for the listed number of episodes.

We consider four different domains from the D4RL benchmark [14] from which **32** different datasets (i.e. tasks) are available. Each dataset here corresponds to a single batch RL task, where we treat the provided data as $\mathcal{D}$, learn a policy $\pi$ using only $\mathcal{D}$, and finally evaluate this policy when it is deployed in the actual environment. In many cases, we have two different datasets taken from the same environment, but collected by behavior policies of varying quality. For example, from the MuJoCo HalfCheetah environment, we have one dataset (HalfCheetah-random) generated under a behavior policy that randomly selects actions and another dataset (HalfCheetah-expert) generated under an expert behavior policy that generates strong returns. Note that our batch RL agents do not have information about the quality of $\pi_b$ since this is often unknown in practice.

The *Gym-MuJoCo* domain consists of four environments (Hopper, HalfCheetah, Walker2d) from which we have 15 datasets built by mixing different behavior policies. Here [14] wanted to examine the effectiveness of a given batch RL method for learning under heterogeneous $\pi_b$. The *FrankaKitchen* domain is based on a 9-degree-of-freedom (DoF) Franka robot in a kitchen environment containing various household items. There are 3 datasets from this environment designed to evaluate the generalization of a given algorithm to unseen states [14]. The *Adroit* domain is based on a 24-DoF simulated robot hand with goals such as: hammering a nail, opening a door, twirling a pen, or picking up and moving a ball. [14] provide 8 datasets from this domain, hoping to study batch RL in settings with small amounts of expert data (human demonstrations) in a high-dimensional robotic manipulation task. Finally, *AntMaze* is a navigation domain based on an 8-DoF Ant quadruped robot, from which the benchmark contains 6 datasets. Here [14] aim to test how well batch RL

agents are able to stitch together pieces of existing observed trajectories to solve a given task (rather than requiring generalization beyond $\mathcal{D}$). Table S6 shows more details about the datasets in our benchmark.

| Domain | Task Name | #Samples | Obs Dims | Action Dims |
|---|---|---|---|---|
| **AntMaze** | antmaze-umaze-v0 | 998573 | 29 | 8 |
| | antmaze-umaze-diverse-v0 | 998882 | 29 | 8 |
| | antmaze-medium-play-v0 | 999092 | 29 | 8 |
| | antmaze-medium-diverse-v0 | 999035 | 29 | 8 |
| | antmaze-large-play-v0 | 999059 | 29 | 8 |
| | antmaze-large-diverse-v0 | 999048 | 29 | 8 |
| **Adroit** | pen-human-v0 | 4950 | 45 | 24 |
| | hammer-human-v0 | 11264 | 46 | 26 |
| | door-human-v0 | 6703 | 39 | 28 |
| | relocate-human-v0 | 9906 | 39 | 30 |
| | pen-cloned-v0 | 495071 | 45 | 24 |
| | hammer-cloned-v0 | 995511 | 46 | 26 |
| | door-cloned-v0 | 995643 | 39 | 28 |
| | relocate-cloned-v0 | 995739 | 39 | 30 |
| **FrankaKitchen** | kitchen-complete-v0 | 3679 | 60 | 9 |
| | kitchen-partial-v0 | 136937 | 60 | 9 |
| | kitchen-mixed-v0 | 136937 | 60 | 9 |
| **Gym-MuJoCo** | halfcheetah-random-v0 | 998999 | 17 | 6 |
| | hopper-random-v0 | 999999 | 11 | 3 |
| | walker2d-random-v0 | 999999 | 17 | 6 |
| | halfcheetah-medium-v0 | 998999 | 17 | 6 |
| | walker2d-medium-v0 | 999874 | 17 | 6 |
| | hopper-medium-v0 | 999981 | 11 | 3 |
| | halfcheetah-medium-expert-v0 | 1997998 | 17 | 6 |
| | walker2d-medium-expert-v0 | 1999179 | 17 | 6 |
| | hopper-medium-expert-v0 | 1199953 | 11 | 3 |
| | halfcheetah-medium-replay-v0 | 100899 | 17 | 6 |
| | walker2d-medium-replay-v0 | 100929 | 17 | 6 |
| | hopper-medium-replay-v0 | 200918 | 11 | 3 |
| | halfcheetah-expert-v0 | 998999 | 17 | 6 |
| | hopper-expert-v0 | 999034 | 11 | 3 |
| | walker2d-expert-v0 | 999304 | 17 | 6 |

Table S6: **Overview of D4RL tasks**. Summary of 32 datasets (i.e. tasks, environments) considered in this work, listing the: domain each dataset stems from, name of each task, number of samples (i.e. transitions) in each dataset, and the dimensionality of the state (**Obs Dims**) and action space (**Action Dims**). To get the numbers listed here, a few samples were omitted from the original datasets using the timeout flag suggested by [14] (Click here for details).

# D  Proofs and Additional Theory

This section contains proofs and details of the constants/assumptions of theories mentioned in the main text.

## D.1  Proof of Theorem 1.

**Theorem 1.** For $\overline{Q}_\theta$ in (5), let $\mathcal{T}_{\text{CDC}} : \overline{Q}_{\theta_t} \to \overline{Q}_{\theta_{t+1}}$ denote the operator corresponding to the $\overline{Q}_\theta$-updates resulting from the $t^{\text{th}}$ iteration of Steps 6-7 of Algorithm 1. $\mathcal{T}_{\text{CDC}}$ is a $L_\infty$ contraction under standard conditions that suffice for the ordinary Bellman operator to be contractive [3, 6, 8, 56]. That is, for any $\overline{Q}_1, \overline{Q}_2$:

$$\sup_{s,a} \left| \mathcal{T}_{\text{CDC}}(\overline{Q}_1(s,a)) - \mathcal{T}_{\text{CDC}}(\overline{Q}_2(s,a)) \right| \leq \gamma \cdot \sup_{s,a} \left| \overline{Q}_1(s,a) - \overline{Q}_2(s,a) \right|$$

In this theorem, we also assume that: $\pi_\phi$ is sufficiently flexible to produce $\arg\max_{\hat{a}} \overline{Q}(s, \hat{a})$ for all $s \in \mathcal{D}$ and the optimization subproblems in Steps 6-7 of Algorithm 1 are solved exactly (ignoring all issues related to function approximation). More formally, we adopt assumptions A1-A9 of Antos et al. [3], although the result from this theorem can also be shown to hold under alternative conditions that suffice for the ordinary Bellman operator to be contractive (see Section 2 of Antos et al. [3]). These assumptions involve regularity conditions on the underlying MDP and the behavior policy, as well as expressiveness restrictions on the hypothesis class of our neural networks.

**Proof** We first consider a simple unpenalized case where $\eta = 0$, $\lambda = 0$, and $M = 1$, i.e. the $Q$-ensemble consists of a single network. By the definition in (5) with $M = 1$: $\overline{Q}_\theta = Q_{\theta_1}$, so Step 6 implements the standard *Bellman-optimality* operator update, when we assume $\pi_\phi$ produces $a = \arg\max_{\hat{a}} \overline{Q}_\theta(s, \hat{a})$. This operator is a contraction under standard conditions [3, 8, 31]. Without this assumption on $\pi_\phi$, Step 6 instead relies on the EMaQ operator, which Theorem 3.1 of Ghasemipour et al. [18] shows is also a contraction for the special case of tabular MDPs.

Next consider $M > 0$ (still with $\eta, \lambda = 0$). Now the target-value $y(s')$ for each single $Q$-network $Q_{\theta_j}$ is determined by $\overline{Q}_\theta$ rather than $Q_{\theta_j}$ alone, i.e. $y(s')$ is given by a convex combination of target networks $\{Q_{\theta_j}\}_{j=1}^M$. By Jensen's inequality and basic properties of convexity, the updates to each $Q_{\theta_j}$ remain a contraction. Therefore the overall update to the convex combination of these networks $\overline{Q}_\theta$ is likewise a contraction.

Next we additionally consider $\eta > 0$. Note that reducing $\Delta_j$ is a non-expansive operation on each $Q_{\theta_j}$, since $\Delta_j(s,a)$ is reduced by shrinking large $\max_{\hat{a}} Q_{\theta_j}(s, \hat{a})$ toward $Q(s,a)$ for the $a$ observed in $\mathcal{D}$ (without modifying $Q_{\theta_j}(s, a')$ for other $a'$). Following the previous arguments, the addition of our $\Delta$ penalty preserves the contractive nature of the $\overline{Q}_\theta$ update.

Finally also consider $\lambda > 0$. In this case, $\pi_\phi$ does not simply concentrate on actions that maximize $\overline{Q}_\theta$, so Step 6 no longer implements the Bellman-optimality operator even with $M = 1, \eta = 0$. However with the likelihood penalty, Step 7 is simply a regularized *policy-improvement* update: With $\eta = 0, M = 1$, Step 6 becomes a *policy-evaluation* calculation where the policy being evaluated is $\tilde{\pi}(a|s) = \arg\max_{\{a_k'\}_{k=1}^N \sim \pi_\phi(\cdot|s')} [\overline{Q}_\theta]$. Since the *Bellman-evaluation* operator is also a contraction under standard conditions [3, 8, 31], our overall argument remains otherwise intact. ∎

## D.2  Proof of Theorem 2.

Theorem 2 (restated below) assures us of the reliability of the policy $\pi_\phi$ produced by CDC, guaranteeing that with high probability $\pi_\phi$ will not have much worse outcomes than the behavior policy $\pi_b$ (where the probability here depends on the size of the dataset $\mathcal{D}$). In batch settings, expecting to learn the optimal policy is futile from limited data. Even ensuring *any* improvement at all over an arbitrary $\pi_b$ is ambitious when we cannot ever test any policies in the environment, and reliability of the learned $\pi_\phi$ is thus a major concern.

**Theorem 2.** Let $\pi_\phi \in \Pi$ be the policy learned by CDC, $\gamma$ denote discount factor, and $n$ denote the sample size of dataset $\mathcal{D}$ generated from $\pi_b$. Also let $J(\pi)$ represent the true expected return

produced by deploying policy $\pi$ in the environment. Under assumptions (A1)-(A4), there exist constants $r^*, C_\lambda, V$ such that with high probability $\geq 1 - \delta$:

$$J(\pi_\phi) \geq J(\pi_b) - \frac{r^*}{(1-\gamma)^2} \sqrt{C_\lambda + \sqrt{(V - \log \delta)/n}}$$

The assumptions adopted for this result are listed below. Similar results can be derived under more general forms of these assumptions, but ours greatly simplify the form of our theorem and its proof.

(A1) The complexity of the function class $\Pi$ of possible policy networks $\pi_\phi$ (in terms of the log-likelihood loss $\log \pi$) is bounded by $V$. Here $V$ is defined as the extension of the VC dimension to real-valued functions with unbounded loss, formally detailed in Section III.D of Vapnik [87]. Similar results hold under alternative complexity measures $V$ from the literature on empirical process theory for density and f-divergence estimation [72, 82, 86].

(A2) Rewards in our environment are bounded such that $r(s, a) \leq r^*$ for all $s \in \mathcal{S}, a \in \mathcal{A}$.

(A3) Our learned Q networks are bounded such that $|Q_\theta(s, a)| < B$ for all $s, a$.

(A4) The likelihoods of our learned $\pi_\phi$ are bounded such that $|\log \pi_\phi(a|s)| < L$ for all $s, a$.

(A5) Each policy update step is carried out using the full dataset rather than mini-batch.

**Proof** Define $r^* = \max\limits_{a,s} |r(s, a)|$, and let $d^{\pi_b}$ denote the marginal distribution of states encountered by acting according to $\pi_b$ starting from the initial state distribution $\mu_0$. Thus $d^{\pi_b}$ describes the probability distribution underlying the states present in our dataset $\mathcal{D}$. Recall the *total variation* distance between probability distributions $p$ and $q$ is defined as: $\text{TV}(p, q) = \int |p(x) - q(x)| \mathrm{d}x$.

From equation (18) in the Proof of Proposition 2 (Appendix A.2) from [88], we have:

$$J(\pi_\phi) \geq J(\pi_b) - \frac{r^*}{(1-\gamma)^2} \mathbb{E}_{s \sim d^{\pi_b}} \left[ \text{TV}\Big(\pi_\phi(\cdot \mid s), \pi_b(\cdot \mid s)\Big) \right]$$

$$\geq J(\pi_b) - \frac{r^*}{\sqrt{2}(1-\gamma)^2} \mathbb{E}_{s \sim d^{\pi_b}} \left[ \sqrt{\text{KL}\Big(\pi_b(\cdot \mid s), \pi_\phi(\cdot \mid s)\Big)} \right]$$

$$\geq J(\pi_b) - \frac{r^*}{\sqrt{2}(1-\gamma)^2} \sqrt{\mathbb{E}_{s \sim d^{\pi_b}} \left[ \text{KL}\Big(\pi_b(\cdot \mid s), \pi_\phi(\cdot \mid s)\Big) \right]}$$

where we used Pinsker's inequality in the second line (c.f. [69]), and the last line is an application of Jensen's inequality for the concave function $f(x) = \sqrt{x}$.

By assumption (A1), each update of our policy network $\pi_\phi$ in CDC is produced via:

$$\phi \leftarrow \underset{\phi}{\text{argmax}} \sum_{(s,a) \in \mathcal{D}, \hat{a} \sim \pi_\phi(\cdot|s)} \left[ \overline{Q}_{\theta_t}(s, \hat{a}) + \lambda \cdot \log \pi_\phi(a|s) \right]$$

where $\theta_t$ denotes the current parameters of our Q networks in iteration $t$ of CDC. Each of these penalized optimizations can be equivalently formulated using a hard constraint, i.e., there exists constant $C_{\lambda, \theta_t} > 0$ (for which $\lambda$ is the corresponding Lagrange multiplier), such that the following optimization leads to the same $\phi$:

$$\phi \leftarrow \underset{\phi}{\text{argmax}} \sum_{(s,a) \in \mathcal{D}, \hat{a} \sim \pi_\phi(\cdot|s)} \left[ \overline{Q}_{\theta_t}(s, \hat{a}) \right]$$
$$\text{subject to } \mathbb{E}_{(s,a) \sim \mathcal{D}} \left[ \log \pi_\phi(a|s) \right] \geq C_{\lambda, \theta_t}$$

**Note:** Throughout, $\mathbb{E}_{(s,a) \sim \mathcal{D}}$ is an *empirical* expectation over dataset $\mathcal{D}$, whereas $\mathbb{E}_{\pi_b}$ denotes true expectations over the underlying distribution of the behavior policy. Since all $Q_{\theta_t}$ are bounded by (A3), so must be

$$C_\lambda^* := \max_t \{C_{\lambda, \theta_t}\} \tag{13}$$

Thus, in every iteration of CDC, the resulting $\pi_\phi$ must satisfy:

$$\mathbb{E}_{s,a \sim \mathcal{D}} \left[ \log \pi_\phi(a|s) \right] \geq C_{\lambda^*} . \tag{14}$$

Finally, we conclude the proof by using Lemma 2 to replace the bound on the empirical likelihood values with a bound on the underlying KL divergence from the data-generating behavior policy distribution. ∎

**Lemma 2** *Suppose* $\mathbb{E}_{(s,a)\sim\mathcal{D}}[\log \pi_\phi(a|s)] \geq C_\lambda^*$. *Then with probability* $\geq 1 - \delta$:

$$\mathbb{E}_{s\sim d^{\pi_b}}\Big[\mathrm{KL}\big(\pi_b(\cdot \mid s), \pi_\phi(\cdot \mid s)\big)\Big] \leq C_\lambda + \sqrt{(V - \log \delta)/n}$$

*where* $n = |\mathcal{D}|$, *and* $d^{\pi_b}$ *denotes the marginal state-visitation distribution under the behavior policy, and:*

$$C_\lambda := C_b - C_\lambda^* \tag{15}$$

*for* $C_\lambda^*$ *defined in* (13) *and constant* $C_b := \mathbb{E}_{s\sim d^{\pi_b}, a\sim\pi_b(\cdot|s)}\big[\log \pi_b(a \mid s)\big]$.

**Proof** A classical result in statistical learning theory (Theorem (23) in Section III.D of Vapnik [87]) states that the following bound simultaneously holds for all $\pi_\phi \in \Pi$ with probability $1 - \delta$:

$$\mathbb{E}_{s\sim d^{\pi_b}, a\sim\pi_b(\cdot|s)}[\log \pi_\phi(a|s)] \geq \mathbb{E}_{(s,a)\sim\mathcal{D}}[\log \pi_\phi(a|s)] - \sqrt{(V - \log \delta)/n} \tag{16}$$

Recall $V$ measures the complexity of function class $\Pi$ (with respect to the log-likelihood loss) and here is defined as the extension of the VC dimension to real-valued functions with unbounded loss from Section III.D of Vapnik [87]. We now write:

$$\begin{aligned}
\mathbb{E}_{s\sim d^{\pi_b}}\Big[\mathrm{KL}\big(\pi_b(\cdot \mid s), \pi_\phi(\cdot \mid s)\big)\Big] &= \mathbb{E}_{s\sim d^{\pi_b}, a\sim\pi_b(\cdot|s)}\big[\log \pi_b(a \mid s)\big] \\
&\quad - \mathbb{E}_{s\sim d^{\pi_b}, a\sim\pi_b(\cdot|s)}\big[\log \pi_\phi(a \mid s)\big] \\
&= C_b - \mathbb{E}_{s\sim d^{\pi_b}, a\sim\pi_b(\cdot|s)}\big[\log \pi_\phi(a \mid s)\big] \\
&\qquad\qquad\qquad\qquad \text{by definition of constant } C_b \\
&\leq C_b - \Big(\mathbb{E}_{(s,a)\sim\mathcal{D}}\big[\log \pi_\phi(a \mid s)\big] - \sqrt{(V - \log \delta)/n}\Big) \\
&\qquad\qquad\qquad\qquad \text{by the empirical process bound in (16)} \\
&\leq C_b - C_\lambda^* + \sqrt{(V - \log \delta)/n} \\
&\qquad\qquad\qquad\qquad \text{from (14)} \\
&= C_\lambda + \sqrt{(V - \log \delta)/n} \\
&\qquad\qquad\qquad\qquad \text{from (15)}
\end{aligned}$$

allowing us to conclude the proof.

∎

### D.3  Proof of Theorem 3.

**Theorem 3:** Define $\mathrm{OE}_{ag}$ as the resultant overestimation bias when performing the maximization step by an agent $ag$: $\mathbb{E}[\max_a Q_\theta(s, a)] - \max_a Q^*(s, a)$. Here $Q_\theta$ denotes the estimate of true Q-value ($Q^*$) learned by ag, which may use **CDC** (with $\eta > 0$) or a **baseline** that uses Algorithm 1 with $\eta = 0$ (with the same value of $\lambda$), and the expectation is taken over the randomness of the underlying dataset, as well as the learning process. Under the assumptions stated below, there co-exist constants $L_1$ and $L_2$ such that

$$\mathrm{OE}_{\mathrm{CDC}} \leq L_1 - \eta L_2 \leq \mathrm{OE}_{\mathrm{baseline}}.$$

This result relies on the following assumptions:

(A1) For a specific state-action pair $\langle s, a_{ID}\rangle$ in our dataset $\mathcal{D}$, we assume a function approximator parameterized by $\theta = \langle \mathbf{q}, V, Q_{ID}\rangle$ as follows:

$$Q_\theta(s, a) = \begin{cases} \sum_{i=1}^m \mathbb{1}\big(a \in \mathcal{A}_i(s)\big)\big(V(s) + q_i\big) & a \in A \setminus a_{ID} \\ Q_{ID} & \text{otherwise} \end{cases}$$

where sets $\mathcal{A}_i \ \forall i \in [1, m]$ form a disjoint non-empty partition of the action space $A$. Here $q_i$ could be thought of as a subset-dependent value, while $V$ is a common (subset-independent) value that quantifies the overall quality of a particular state $s$, also considered by Wang et al. [59]. Note that such discretization schemes are commonly assumed in the RL literature [83], and that such function approximators can closely approximate all continuous functions up to any desired accuracy as $m$ increases. At the cost of a more complex analysis, we may have alternatively assumed Lipschitz continuous function approximators, since they are closely related to discretization through the notion of covering number.

(A2) Following Thrun and Schwartz [58][5], we study overestimation in a particular state $s$ with $\forall a \ Q^*(s, a) = C$. Previous work studied this case because it is the setting where maximal overestimation occurs (due to maximal Q-value overlap). Stated differently, not assuming A2 means our theorem still provides the following bound: $\max(\text{OE}_{\text{CDC}}) \leq \max(\text{OE}_{\text{baseline}})$ where the maximum is taken over all possible true Q functions.

(A3) We assume $\forall s, i :$ the $q_i$ are independent and are distributed uniformly in $[-L_1, L_1]$. This assumption is also adopted from the literature [32, 58] and greatly simplifies our analysis. We further assume $Q_\theta(s, a_{ID}) = Q_{ID}$, where $Q_{ID}$ is distributed uniformly at random in $[-\alpha L_1, \alpha L_1]$ for $\alpha \ll 1$, reflecting the conviction that our $Q$ estimates should generally be more accurate for the previously observed state-action pairs.

(A4) We assume that Step 6 of Algorithm 1 proceeds by first updating $\theta$ based on taking a gradient-step towards minimizing $\eta \Delta(s, a)$ term (see (6)): $\theta \leftarrow \theta - \mu \eta \nabla_\theta \Delta(s, a)$, followed by performing the maximization step in minimizing the TD error. Separating the update into two steps simplifies our analysis while remaining to be a reasonable way to perform Step 6 of Algorithm 1. Here, $\mu$ is the step-size.

(A5) We assume the policy $\pi_\phi$ assigns non-zero probability to at least one action from each $\mathcal{A}_i$.

**Proof** We begin by quantifying the overestimation bias for the baseline case. By denoting $a_i$ to be a member of $\mathcal{A}_i(s)$, and in light of our function approximator (assumption A1), we can write:

$$
\mathbb{E}\Big[ \max_{a \in \mathcal{A}} Q_\theta(s, a) \Big] = \mathbb{E}\Big[ \max \big\{ Q_\theta(s, a_{ID}), Q_\theta(s, a_1), \cdots, Q_\theta(s, a_m) \big\} \Big]
$$

$$
\text{(assumptions A2, and A3)}
$$

$$
= \mathbb{E}\Big[ \max\{ Q_{ID}, C + q_1, \cdots, C + q_m \} \Big]
$$

$$
\text{(assumption A3)}
$$

$$
\geq \mathbb{E}\Big[ \max \big\{ C - \alpha L_1, C + q_1, \cdots, C + q_m \big\} \Big]
$$

$$
= C + \mathbb{E}\Big[ \max \big\{ -\alpha L_1, q_1 - \alpha L_1 + \alpha L_1, \cdots, q_m - \alpha L_1 + \alpha L_1 \big\} \Big]
$$

$$
= C - \alpha L_1 + \mathbb{E}\Big[ \max \big\{ 0, q_1 + \alpha L_1, \cdots, q_m + \alpha L_1 \big\} \Big]
$$

$$
= C - \alpha L_1 + m \int_{x:-\alpha L_1}^{L_1} \frac{(x + \alpha L_1)}{2L_1} \Big( \int_{y:-L_1}^{x} \frac{1}{2L_1} dy \Big)^{m-1} dx
$$

$$
= C - \alpha L_1 + m \int_{x:-\alpha L_1}^{L_1} \frac{(x + \alpha L_1)}{2L_1} \Big( \frac{x + L_1}{2L_1} \Big)^{m-1} dx \ .
$$

In the penultimate step, the equality holds because we only need to consider cases where at least one of the random variables is bigger than $-\alpha L_1$, because otherwise the maximum is 0, thus not affecting the expectation. We broke down the expectation to $m$ cases, where in each case the maximizing noise is at least $-\alpha L_1$ (the integral over $x$), and the remaining $n - 1$ variables are smaller than the maximizing one (the integral over $y$).

Using a change-of-variable technique ($z = \frac{x + L_1}{2L_1}$), we can then write:

---

[5]Another paper with this assumption is the work of Lan et al. [32] who did not state this assumption, but to get their second equality on their page 12, the assumption A2 needs to hold.

$$\mathbb{E}\Big[\max_{a\in\mathcal{A}}Q_\theta(s,a)\Big] \geq C - \alpha L_1 + m\int_{x:-\alpha L_1}^{L_1}\frac{(x+\alpha L_1)}{2L_1}\Big(\frac{x+L_1}{2L_1}\Big)^{m-1}dx$$

$$= C - \alpha L_1 + mL_1\int_{y:\frac{1-\alpha}{2}}^{1}\big(2y-(1-\alpha)\big)y^{m-1}dy$$

$$= C - \alpha L_1 + \underbrace{mL_1\Big(\frac{2}{m+1}-\frac{1-\alpha}{m}\Big) + L_1\Big(\frac{(\frac{1-\alpha}{2})^{m+1}}{m+1}\Big)}_{:=f(L_1,m,\alpha)}, \qquad (17)$$

allowing us to write that: $\mathrm{OE_{baseline}} \geq -\alpha L_1 + f(L_1,m,\alpha)$. Notice that $\lim_{m\to\infty}f(L_1,m,\alpha) = L_1 + \alpha L_1$. For example, with $\alpha = 0$ (for which the bound is tight), the overestimation bias of the baseline for $m = 2$ is $C + f(2,L_1,0) - \max_a Q(s,a) = C + f(2,L_1,0) - C = f(2,L_1,0) = \frac{5L_1}{12}$ and monotonically increases and converges to $L_1$ as $m \to \infty$.

We now move to the case where we perform the CDC update prior to the maximization step (assumption A4). The update proceeds by first choosing the maximum OOD action, which given assumption A5 corresponds to: $\max_{i\in[1,m]}Q_\theta(s,a_i)$.

Now define $\varepsilon_+ := \max\big\{0, \max_i Q_\theta(s,a_i) - Q_\theta(s,a_{ID})\big\}$, we have:

$$\mathbb{E}\Big[\varepsilon_+\Big] = \mathbb{E}\Big[\max\big\{0, \max_i Q_\theta(s,a_i) - Q_\theta(s,a_{ID})\big\}\Big]$$

$$= \mathbb{E}\Big[\max\big\{0, \max_i Q_\theta(s,a_i) - (Q_{ID})\big\}\Big]$$

$$\text{(from Assumption A3, namely: } Q_{ID} \leq C + \alpha L_1)$$

$$\geq \mathbb{E}\Big[\max\big\{0, \max_i Q_\theta(s,a_i) - C - \alpha L_1\big\}\Big]$$

$$= \mathbb{E}\Big[\max\big\{0, \max_i C + q_i - C - \alpha L_1\big\}\Big] = \mathbb{E}\Big[\max\big\{0, \max_i q_i - \alpha L_1\big\}\Big]$$

$$= \mathbb{E}\Big[\max\big\{2\alpha L_1, \max_i q_i + \alpha L_1\big\}\Big] - 2\alpha L_1$$

$$\geq \mathbb{E}\Big[\max\big\{0, \max_i q_i + \alpha L_1\big\}\Big] - 2\alpha L_1$$

$$= \mathbb{E}\Big[\max\big\{0, q_1 + \alpha L_1, \cdots, q_m + \alpha L_1\big\}\Big] - 2\alpha L_1$$

$$\text{(from our analysis of the baseline case and (17))}$$

$$= f(L_1,m,\alpha) - 2\alpha L_1 \ .$$

Without loss of generality, assume that the maximizing action is $a_1 = \mathrm{argmax}_a Q_\theta(s,a)$. Thus we can quantify the expected CDC update as follows (see also (8)):

$$\Big(\boldsymbol{\nabla}_\theta Q_\theta(s,a_1) - \boldsymbol{\nabla}_\theta Q_\theta(s,a_{ID})\Big)\mathbb{E}\Big[\varepsilon_+\Big] \geq \Big(\boldsymbol{\nabla}Q_\theta(s,a_1) - \boldsymbol{\nabla}Q_\theta(s,a_{ID})\Big)\Big(f(L_1,m,\alpha) - 2\alpha L_1\Big) \tag{18}$$

Given the form of our function approximator, by performing the updates (assumption A4)

$$\begin{bmatrix}\mathbf{q}\\V(s)\\Q_{ID}\end{bmatrix} \leftarrow \begin{bmatrix}\mathbf{q}\\V(s)\\Q_{ID}\end{bmatrix} - \mu\eta\Big[\Big(\boldsymbol{\nabla}_\theta Q_\theta(s,a_1) - \boldsymbol{\nabla}_\theta Q_\theta(s,a_{ID})\Big)\varepsilon_+\Big],$$

CDC will:

- inflate the value of $Q_\theta(s,a_{ID})$ by at least $\mu\eta\Big(f(L_1,m,\alpha) - 2\alpha L_1\Big)$. This is due to updating $Q_{ID}$.

- deflate $Q_\theta(s,a_1)$ by at least $2\mu\eta\Big(f(L_1,m,\alpha) - 2\alpha L_1\Big)$. This is due to updating $q_1$ and $V(s)$.

- deflate $Q_\theta(s, a_i)$ $\forall i \neq 1$ by at least $\mu\eta\Big(f(L_1, m, \alpha) - 2\alpha L_1\Big)$, due to updating $V(s)$.

Now notice that, based on (18), we will at least subtract $\eta\mu\Big(f(L_1, m, \alpha) - 2\alpha L_1\Big)$ from all $Q_\theta(s, a)$, whose value could at most have been $C + L_1$ prior to the update (assumption A3). Therefore we can claim:

$$\mathbb{E}\Big[\max_a Q_\theta(s, a)\Big] \leq C + L_1 - \eta \underbrace{\mu\Big(f(L_1, m, \alpha) - 2\alpha L_1\Big)}_{L_2},$$

and that:

$$\mathbb{E}\Big[\text{OE}_{\text{CDC}}\Big] = \mathbb{E}\Big[\max_a Q_\theta(s, a)\Big] - \max_a Q^*(s, a) \leq C + L_1 - \eta L_2 - C = L_1 - \eta L_2$$

Further, for $\text{OE}_{\text{CDC}} \leq \text{OE}_{\text{baseline}}$ to hold, we need that:

$$f(L_1, m, \alpha) \geq \frac{(1 + 2\alpha\eta\mu + \alpha)L_1}{1 + \mu\eta},$$

For sufficiently small $\mu$, i.e. $\mu < \frac{1-\alpha}{2\alpha\eta}$, and sufficiently large $m$, i.e. $m \geq \frac{1}{1/2 - \frac{1}{1+\eta\mu}}$, we get the desired result: $\text{OE}_{\text{CDC}} \leq \text{OE}_{\text{baseline}}$, allowing us to conclude the proof. ∎

## Additional References for the Appendix

[1] R. Agarwal, D. Schuurmans, and M. Norouzi. An optimistic perspective on offline reinforcement learning. In *International Conference on Machine Learning*, 2020.

[3] A. Antos, R. Munos, and C. Szepesvari. Fitted Q-iteration in continuous action-space MDPs. In *Advances in Neural Information Processing Systems*, 2007.

[6] D. P. Bertsekas and S. Shreve. *Stochastic optimal control: the discrete-time case*. Athena Scientific, 2004.

[8] L. Busoniu, R. Babuska, B. De Schutter, and D. Ernst. *Reinforcement learning and dynamic programming using function approximators*, volume 39. CRC press, 2010.

[69] I. Csiszár and J. Körner. *Information Theory: Coding Theorems for Discrete Memoryless Systems*. Cambridge University Press, 2 edition, 2011. doi: 10.1017/CBO9780511921889.

[14] J. Fu, A. Kumar, O. Nachum, G. Tucker, and S. Levine. D4rl: Datasets for deep data-driven reinforcement learning. *arXiv:2004.07219*, 2020.

[17] S. Fujimoto, D. Meger, and D. Precup. Off-policy deep reinforcement learning without exploration. In *International Conference on Machine Learning*, pages 2052–2062, 2019.

[72] S. A. Geer and S. van de Geer. *Empirical Processes in M-estimation*, volume 6. Cambridge university press, 2000.

[18] S. K. S. Ghasemipour, D. Schuurmans, and S. S. Gu. Emaq: Expected-max q-learning operator for simple yet effective offline and online rl. *arXiv:2007.11091*, 2021.

[20] C. Gulcehre, S. G. Colmenarejo, Z. Wang, J. Sygnowski, T. Paine, K. Zolna, Y. Chen, M. Hoffman, R. Pascanu, and N. de Freitas. Regularized behavior value estimation, 2021.

[21] T. Haarnoja, A. Zhou, P. Abbeel, and S. Levine. Soft actor-critic: Off-policy maximum entropy deep reinforcement learning with a stochastic actor. *arXiv:1801.01290*, 2018.

[28] A. Kumar, J. Fu, G. Tucker, and S. Levine. Stabilizing Off-Policy Q-Learning via Bootstrapping Error Reduction. *arXiv:1906.00949*, Nov. 2019.

[29] A. Kumar, A. Zhou, G. Tucker, and S. Levine. Conservative Q-Learning for Offline Reinforcement Learning. *arXiv:2006.04779*, June 2020.

[31] M. G. Lagoudakis and R. Parr. Least-squares policy iteration. *The Journal of Machine Learning Research*, 4:1107–1149, 2003.

[32] Q. Lan, Y. Pan, A. Fyshe, and M. White. Maxmin q-learning: Controlling the estimation bias of q-learning. In *International Conference on Learning Representations*, 2019.

[35] H. Le, C. Voloshin, and Y. Yue. Batch policy learning under constraints. In *Proceedings of the 36th International Conference on Machine Learning*, volume 97 of *Proceedings of Machine Learning Research*, pages 3703–3712. PMLR, 2019.

[41] V. Mnih, K. Kavukcuoglu, D. Silver, A. A. Rusu, J. Veness, M. G. Bellemare, A. Graves, M. Riedmiller, A. K. Fidjeland, G. Ostrovski, et al. Human-level control through deep reinforcement learning. *Nature*, 518(7540):529–533, 2015.

[82] X. Nguyen, M. J. Wainwright, and M. I. Jordan. Estimating divergence functionals and the likelihood ratio by convex risk minimization. *IEEE Transactions on Information Theory*, 56 (11):5847–5861, 2010.

[83] R. S. Sutton and A. G. Barto. *Reinforcement learning: An introduction*. MIT press, 2018.

[56] C. Szepesvári. Efficient approximate planning in continuous space markovian decision problems. *AI Communications*, 14(3):163–176, 2001.

[58] S. Thrun and A. Schwartz. Issues in using function approximation for reinforcement learning. In *Proceedings of the 1993 Connectionist Models Summer School*, pages 255–263. Lawrence Erlbaum, 1993.

[86] A. van der Vaart, A. van der Vaart, A. W. van der Vaart, and J. Wellner. *Weak Convergence and Empirical Processes: With Applications to Statistics*. Springer Science & Business Media, 1996.

[87] V. N. Vapnik. An overview of statistical learning theory. *IEEE transactions on neural networks*, 10(5):988–999, 1999.

[88] Q. Wang, J. Xiong, L. Han, P. Sun, H. Liu, and T. Zhang. Exponentially weighted imitation learning for batched historical data. In *Advances in Neural Information Processing Systems*, volume 31. Curran Associates, Inc., 2018.

[61] F. Wilcoxon. Individual comparisons by ranking methods. *Biometrics Bulletin*, 1(6):80–83, 1945.

[63] Y. Wu, G. Tucker, and O. Nachum. Behavior Regularized Offline Reinforcement Learning. *arXiv:1911.11361*, 2019.