# OpenReview forum: "Continuous Doubly Constrained Batch Reinforcement Learning"
_NeurIPS.cc/2021/Conference — NeurIPS 2021 Poster_

### Official Review · Reviewer_jjjy · 2021-06-24

**Rating:** 6
**Confidence:** 3

**Summary:**

The paper proposes a new algorithm for batch Reinforcement Learning, which differs from existing solutions by the addition of an additional term to the critic loss, aimed at penalising overestimation as well as adding a behavior-cloning-style term to the actor loss, aiming at keeping generated actions close to the distribution of data in the batch.

**Ethical Concerns:**

I did not identify ethical issues with this submission.

**Limitations And Societal Impact:**

In the checklist, the authors say that assessment of societal impact does not apply. It is true that the impact of this RL algorithm is similar to others measured against the same benchmarks, but I feel an assessment should have been included. One ethics-related aspect of batch RL that I think merits discussion would be its ability to learn policies interacting with humans without involving actual human subjects (which can be used both for good and bad purposes).

**Main Review:**

The is a good fit for the NeurIPS community in terms of scope. It shows good empirical results. The paper is clearly written, although there are some minor omissions (see “minor points” below).

My main concerns are the following.
1. The novelty of the contribution isn't huge - the paper is an incremental extension of previous techniques.
2. Since the paper seems to be mainly an empirical contribution, the ability to reproduce the results is critical. However, the authors do not provide the source code and the information provided in the appendix is limited to pseudocode and a list of hyperparameters.
3. The paper discusses both forward and reverse KL for the policy regularization but only uses the reverse KL. It is not clear what the benefit of including the forward KL in Lemma 1 is (it makes the paper harder to read because one is kept wondering when it will be used).
4. The proofs are a bit too informal. For example, line 774 in the appendix reads “From standard empirical process theory used throughout statistical learning [85], we have that the following bound simultaneously holds”. This does not really qualify as including a full proof as per 2b in the checklist. Is is OK to reference standards results from sources external to the paper, but they should be fully identified (i.e. Lemma [number] by [other author]). This makes the math difficult to follow or check. Overall, I think the paper could benefit from making the theoretical results less general, but more complete.

Overall, due to the strength of the empirical results, despite these concerns, I score this paper as a weak accept.

Minor points:
- line 76: is -> are;
- line 112: EMAQ -> EMaQ;
- line 121: non-OOD -> in-distribution;
- Eq 10: the \propto sign is used in an unconventional way (it normally denotes scaling);
- line 297: learned by ag (what does it mean?);
- Bibliography: [31] and [32] are the same paper.

**Time Spent Reviewing:**

4

---

> ### Author Response · Authors · 2021-08-10
> **Response to Reviewer jjjy**
>
> Thank you for your valuable feedback.
>
>
> > authors do not provide the source code and the information provided in the appendix is limited to pseudocode and a list of hyperparameters.
>
> We will release the code for CDC with the final version of the paper. That being said, our CDC implementation is exactly as described in the paper (Algorithm 1) and we try to be as transparent as possible about all implementation details. We also provided evaluation code in Algorithm 3 and information about computing infrastructure and software libraries in Table S4 to make it easier to implement CDC while its code is not available.
>
>
> >  I think the paper could benefit from making the theoretical results less general, but more complete. The proofs are a bit too informal. For example, line 774 in the appendix reads “From standard empirical process theory used throughout statistical learning [85], we have that the following bound simultaneously holds”. This does not really qualify as including a full proof as per 2b in the checklist.
>
> We agree with the suggestion and have restated some of our theorems’ assumptions/proofs more rigorously. Some of the particular changes include:
>
> We have rewritten line 774 to be more precise:
>
> A classic result in statistical learning theory, Theorem (23) in Section D of (Vapnik, 1999: https://www.math.arizona.edu/~hzhang/math574m/Read/vapnik.pdf), states that the following bound holds:
> …
> Here constant $V$ measures the complexity of function class $\Pi$ with respect to the log-likelihood loss, and is defined as the extension of the VC dimension to real-valued functions with unbounded loss formally described in Section D of (Vapnik, 1999).
>
> And we’ve accordingly rewritten assumption (A1) for Theorem 2 as well:
>
> (A1) The complexity of the function class $\Pi$ of possible policy networks $\pi_\phi$ (in terms of the log-likelihood loss $log \pi$) is bounded by $V$. Here $V$ measures complexity via the . Similar results may be obtained under more sophisticated complexity measures $V$ discussed in the literature on empirical process theory for density and f-divergence estimation [72, 80, 84].
>
>
> We have now added explicit reference to formal statements of the assumptions required for Theorem 1, as these are quite involved to explain in this paper. The newly added sentence to Theorem 1 restatement in Appendix reads as follows:
>
> Formally, we adopt assumptions A1-A9 of Antos et al. (2007: https://papers.nips.cc/paper/2007/file/da0d1111d2dc5d489242e60ebcbaf988-Paper.pdf), although the result from this theorem can also be shown to hold under alternative conditions that suffice for the ordinary Bellman operator to be contractive (see discussion in Section 2 of Antos et al. (2007)). These assumptions involve regularity conditions on the underlying MDP and the behavior policy, as well as expressiveness restrictions on the hypothesis class of our neural networks (similar assumptions are also generally needed for theory related to supervised learning).
>
> Corresponding lines of the proofs have also been rewritten in this more precise/rigorous style where appropriate.
>
>
>
> > line 297: learned by ag (what does it mean?);
>
> The shorthand “ag” was just used to make it easier to refer to estimates obtained by an RL agent who either runs CDC or some other batch RL algorithm, but clearly this was confusing notation.  This is easily improved and we have replaced this notation in the final version to be much clearer.
>
> > Minor points:
>
> Thanks for pointing out those typos and minor writing issues. We have addressed them in the final version of the paper.
>
>
> > One ethics-related aspect of batch RL that I think merits discussion would be its ability to learn policies interacting with humans without involving actual human subjects (which can be used both for good and bad purposes).
>
> Good point, we have added a concise discussion of societal impact:
>
> A big advantage of batch RL is that it can be applied to learn policies interacting with humans without subjecting human subjects to harm from exploratory actions as in online experimentation. However this potential benefit comes with many caveats. In particular, it is critical that the offline dataset has been collected without selection bias in order for batch RL to perform reliably (see the causal inference literature for further discussions on the difficulty of dealing with such latent confounding). This can only be ensured through domain expertise and deep understanding of the application in which the data are collected to recognize potential pitfalls. Difficult examples of selection bias that can arise even with methodical data curation include a population that has evolved over time since the data were collected, or a population whose behavior changes in response to deployment of the learned policy. Another issue in high-stakes applications is the need for uncertainty quantification regarding how sure we are that the offline-learned policy is actually superior to the behavior policy, such that “do no harm” tradeoffs can be appropriately considered when deciding whether to upgrade an existing policy to the learned one. Unfortunately, even with the improvements gained by incorporating our Q-value regularization into off-policy evaluation, today’s OPE estimates remain quite inaccurate on challenging benchmarks like D4RL. This further highlights the scientific need for not only better OPE estimators, but also researching suitable uncertainty estimates that can at least quantify this potential inaccuracy.

---

> > ### Comment · Reviewer_jjjy · 2021-08-18
> > **Thanks for the author repsonse.**
> >
> > Thanks for responding to my points, making the theory more concrete and adding a discussion of societal impact.

---

> > > ### Author Response · Authors · 2021-08-18
> > > **Thanks.**
> > >
> > > Thanks again for your feedback and comments. We are glad that our responses address your concerns and we hope you will consider increasing your score. Please let us know if you have more questions.

---

### Official Review · Reviewer_T9dA · 2021-07-13

**Rating:** 7
**Confidence:** 4

**Summary:**

The paper tackles the classical problem of batch RL. As a starting point it takes an actor-critic method with Q-function, plus EMAQ Q update and low-confidence bounds with ensembles of Q functions. Authors introduce two methods for regularizing the model:
- Extra-overestimation penalty, which discourages, for a given state, actions not present in the dataset to have higher Q-value than the actions present there
- Exploration penalty, encouraging high KL between behavioral and learned policy

The resulting method is tested on the standard benchmark: D4RL and the results compared against a number of contemporary batch RL methods.

**Limitations And Societal Impact:**

Yes

**Main Review:**

1. The paper is written very clearly, and authors provide convincing intuitions behind the methods they propose.
2. The experimental section is impressive: authors compared their method against a number of baselines and provided reasonable ablations of their work.
3. The results themselves are not groundbreaking, but good enough for an accepted paper.
4. Authors provide theoretical guarantees (Theorem 2, 3) about their method which feel strong. I haven't been able to verify they are correct, but they seem theoretically plausible (rewards not much worse than behavioral policy, with the bound tightening the stronger the regularization and the bigger the dataset).
5. In Theorem 1, authors claim that their Q-update operator T is a contraction and Q-function converges under "commonly-assumed conditions". It's not clear to me what these are; the readers would appreciate it if authors stated them explicitly, at least in the Appendix. Assuming that the authors mean the conditions mentioned in sec. 2.3.2 of Busoniu et al. (2010), they require that "all the state-action pairs are (asymptotically) visited infinitely often", which is by definition not the case for batch RL, making the Theorem 1 useless.
6. Authors fail to cite a relevant work: Gulcehre et al. ["Addressing Extrapolation Error in Deep Offline Reinforcement Learning"](https://openreview.net/forum?id=OCRKCul3eKN), where a very similar regularization term:
$$\sum_{\hat{a}\in{a_1,\ldots, a_N}}[Q(s, \hat{a}) - Q(s, a) - v]_+^2$$
is used in the same context and with the same motivation as the authors' extra-overestimation term. I think it would be appropriate to refer to this work, and ideally compare to it experimentally, especially given a similar structure to the paper (a regularizer for Q, another method for keeping the policy close to the behavior one). It seems one could mix&match between the parts proposed by these two papers and it may be interesting to see results of such an ablation.

Overall, I find the paper sound, and suggest acceptance.

Smaller comments:

- In 182, $\varepsilon$ is not defined. Is it meant to be $Q(s, \hat{a}) - Q(s, a)$ ?
- EMAQ is written inconsistently in 91 vs 112, 266


**Time Spent Reviewing:**

3

---

> ### Author Response · Authors · 2021-08-10
> **Response to Reviewer T9dA**
>
> Thank you for your valuable feedback and suggested experiments.
>
> >  cite a relevant work: Gulcehre et al.
>
> Thank you for pointing out the related work of Gulcehre et al. (Official title on Arxiv: “Regularized Behavior Value Estimation” https://arxiv.org/pdf/2103.09575.pdf ). We were unaware of this work as it was only posted to Arxiv in March 2021, two months before the Neurips submission deadline. We have added a discussion comparing our work with Gulcehre et al. to our revised paper:
>
> Gulchehre et al. concurrently propose a value regularization term for batch RL that is similar to the Q-value regularizer used by CDC.  Unlike CDC, Gulchehre et al. only consider discrete actions under a DQN framework rather than the actor-critic RL framework employed in CDC.  Gulchehre et al. do not consider explicit policy regularization (as they use DQN without explicitly representing a policy), which forms a critical component of CDC to supplement its value regularization.  Our ablation studies in Figure 2a and Table S2 demonstrate that policy regularization is critical required to achieve good performance in CDC, and value regularization alone is insufficient.  That said, Gulchehre et al. do also acknowledge the importance of ensuring the learned policy does not stray too far from the behavior policy, but their proposal to ensure this involves restricting the learner to apply only a single step of policy-iteration to the estimated value function. However in continuous action spaces with a policy-network, even a single policy-iteration step can lead to large deviations from the behavior policy without explicit policy regularization as imposed by CDC (how much deviation can occur within one policy-iteration step will entirely depend on properties of the Q-estimates).
>
> Finally, we note that the methodology of Gulchehre et al. requires a dataset that contains observations (s, a, r, s’, a’), i.e. more complete subtrajectories of episodes, whereas CDC merely requires a dataset that contains observations of the form (s, a, r, s’). The former setting is less widely applicable, but is somewhat easier due to the availability of the subsequent action a’ for temporal-difference learning.
>
>
> >  compare CDC with Gulcehre et al.
>
> We have run the RBVE method of Gulchehre et al. on our D4RL benchmark, after first minorly adapting it to the setting considered in our paper. The differences in our setting are: (i) we have continuous actions, and (ii) a’ is not contained in the dataset. In our adaptation, we approximate the max_a required by Gulchehre et al. (but which is difficult for continuous actions) by sampling many actions and taking the empirical maximum. Our adaptation accounts for the fact that a’ is not present in the dataset by first estimating the behavior policy pi_b via behavior-cloning (via maximum likelihood with our same policy network) and then drawing a’ ~ pi_b( . | s’) for use in the method of Gulchehre et al. In addition, we ran two sets of experiments. In the first experiment (called RBVE-A), soft filtering weight, $w(s,a)$, is implemented according to Eq 6 in their paper. In the second experiment (called RBVE-C), we treat $\omega$ as a hyperparameter and we use a fixed value that worked best in a sweep. Since soft filtering weights didn’t perform well in our experiments, we decided to do the second set of runs to ensure that we are being thorough in these experiments and considering both cases.
>
> The following table shows the results of these experiments. These results clearly demonstrate that CDC outperforms both versions of RBVE. Note that we use the exact same setup for CDC as before. In addition, we show learning curves here: [ https://imgur.com/a/NBQBTke ].
>
> Task Name  RBVE-C  RBVE-A  CDC
>
>
> halfcheetah-random              |	18.89|	-0.01|	$\mathbf{27.36}$
>
>
> halfcheetah-medium              |	43.98|	24.27|	$\mathbf{46.05}$
>
>
> halfcheetah-medium-replay       |	37.24|	7.55|	$\mathbf{44.74}$
>
>
> halfcheetah-medium-expert       |	32.11|	8.12|	$\mathbf{59.64}$
>
>
> halfcheetah-expert              |	36.57|	3.14|	$\mathbf{82.05}$
>
>
> hopper-random                   |	11.46|	4.69|	$\mathbf{14.76}$
>
>
> hopper-medium                   |	17.16|	1.23|	$\mathbf{60.39}$
>
>
> hopper-medium-replay            |	28.15|	3.31|	$\mathbf{55.89}$
>
>
> hopper-medium-expert            |	$\mathbf{88.72}$|	3.62|	86.9
>
>
> hopper-expert                   |	94.32|	1.2|	$\mathbf{102.75}$
>
>
> walker2d-random                 |	0.36|	2.4|	$\mathbf{7.22}$
>
>
> walker2d-medium                 |	80.19|	2.81|	$\mathbf{82.13}$
>
>
> walker2d-medium-replay          |	6.7|	2.41|	$\mathbf{22.96}$
>
>
> walker2d-medium-expert          |	$\mathbf{77.79}$|	1.83|	70.91
>
>
> walker2d-expert                 |	60.3|	0.69|	$\mathbf{87.54}$
>
>
> antmaze-umaze                   |	0.0|	0.0|	$\mathbf{91.85}$
>
>
> antmaze-umaze-diverse           |	2.96|	0.0|	$\mathbf{62.59}$
>
>
> antmaze-medium-play             |	0.0|	0.0|	$\mathbf{55.19}$
>
>
> antmaze-medium-diverse          |	0.0|	0.0|	$\mathbf{40.74}$
>
>
> antmaze-large-play              |	0.0|	0.0|	$\mathbf{5.19}$
>
>
> antmaze-large-diverse           |	0.0|	0.0|	$\mathbf{11.85}$
>
>
> pen-human                       |	24.04|	34.93|	$\mathbf{73.19}$
>
>
> pen-cloned                      |	42.86|	-0.39|	$\mathbf{49.18}$
>
>
> hammer-human                    |	0.59|	0.0|	$\mathbf{4.34}$
>
>
> hammer-cloned                   |	0.34|	0.17|	$\mathbf{2.37}$
>
>
> door-human                      |	$\mathbf{9.15}$|	-0.0|	4.62
>
>
> door-cloned                     |	$\mathbf{0.04}$|	0.03|	0.01
>
>
> relocate-human                  |	0.29|	0.01|	$\mathbf{0.73}$
>
>
> relocate-cloned                 |	-0.23|	$\mathbf{0.01}$|	-0.24
>
>
> kitchen-complete                |	18.61|	0.83|	$\mathbf{58.7}$
>
>
> kitchen-partial                 |	8.7|	0.83|	$\mathbf{42.5}$
>
>
> kitchen-mixed                   |	5.93|	1.11|	$\mathbf{42.87}$
>
>
>
> Total Score 	 |747.21 	 |104.79 	 |$\mathbf{1396.99}$
>
>
> > In Theorem 1, authors claim that their Q-update operator T is a contraction and Q-function converges under "commonly-assumed conditions". It's not clear to me what these are; the readers would appreciate it if authors stated them explicitly, at least in the Appendix…
>
>
> Note that the "all the state-action pairs are (asymptotically) visited infinitely often" assumption in sec 2.3.2 of Busoniu et al. is only required for them to prove that “Q-learning asymptotically converges to Q*”.  In contrast, we are merely proving that CDC’s Q-update is a contraction, and thus do not require this assumption. In fact, it is impossible to provably recover Q* in a batch RL setting with limited data (where we do not sufficiently observe all state-action pairs), exactly as the reviewer points out.
>
> We have now added explicit reference to formal statements of the assumptions in the Appendix, as these are quite involved to explain in this paper. The newly added sentence to Theorem 1 restatement in Appendix reads as follows:
>
> Formally, we adopt assumptions A1-A9 of Antos et al. ( 2007: https://papers.nips.cc/paper/2007/file/da0d1111d2dc5d489242e60ebcbaf988-Paper.pdf ), although the result from this theorem can also be shown to hold under alternative conditions that suffice for the ordinary Bellman operator to be contractive (see discussion in Section 2 of Antos et al. (2007)). These assumptions involve regularity conditions on the underlying MDP and the behavior policy, as well as expressiveness restrictions on the hypothesis class of our neural networks (similar assumptions are also generally needed for theory related to supervised learning).
>
>
> >  In 182, $\varepsilon$ is not defined. Is it meant to be $Q(s, \hat{a}) -  Q(s, a)$?
>
> Yes, you are right: ${\varepsilon}$ is $Q_{\theta}(s, \hat{a}) -  Q_{\theta}(s, a)$. It is already defined in line 184, but we make sure it is more clearly referenced in the final version. Thanks.
>
>
> > EMAQ is written inconsistently
>
> Thanks for pointing out this typo. We have addressed this in the revised paper.

---

> > ### Comment · Reviewer_T9dA · 2021-08-11
> > **Thanks for the answer**
> >
> > Thank you for clarifying the differences to Gulcehre et al., running a comparison, and clarifying the assumptions.

---

> > > ### Author Response · Authors · 2021-08-16
> > > **Thanks.**
> > >
> > > Thanks again for suggesting those experiments, comments, and responses. Please let us know if you have more questions.

---

### Official Review · Reviewer_JguC · 2021-07-14

**Rating:** 6
**Confidence:** 4

**Summary:**

This paper introduces two batch-RL regularizers in order to solve the wild exploration and overestimation problem. The experiments on 32 tasks on D4RL demonstrate that the proposed method is better than the majority of batch-RL methods.

**Limitations And Societal Impact:**

The authors have addressed all of them.

**Main Review:**

$\textbf{Originality}$: The method is new and it innovatively proposes three principles in regularizing Q values. However, the framework of CDC is still similar to that of BRAC, regularization on both value and policy.

$\textbf{Quality}$: The overall quality of the paper is satisfying, there are solid experimental results and statistical tests supporting the advantages of CDC. The theoretical foundation of CDC is also provided.

$\textbf{Clarity}$: I'm confusing about the relation between the two regularizers proposed in the paper, are they independent components? If one of the regularizers is replaced by the technique used in other offline RL algorithms, how will the overall performance change? I think the authors need to clarify this.

$\textbf{Significance}$: The performance of CDC is shown to be significantly better than other offline RL algorithms and the ablation study demonstrates the necessity of each component. The design principles of the Q value regularizer are meaningful but I'm unclear about the motivation of the policy regularizer. In BRAC dual format, the policy regularizer is also free of the estimation of behavior policy $\pi_{b}$, I think one ablation study is needed to show the advantage of the policy regularizer in CDC

**Time Spent Reviewing:**

4

---

> ### Author Response · Authors · 2021-08-10
> **Response to Reviewer JguC**
>
> Thank you for your valuable feedback.
>
>
>  > CDC is still similar to that of BRAC, regularization on both value and policy.
>
> Although BRAC adds regularization to both value and policy updates, both BRAC penalties are intended to regularize the learned policy $\pi_\theta$ towards the behavior policy $\pi_b$ and thus both  are actually policy constraints (see Eq 6 and Eq 7 in BRAC paper). While that is fair to say the BRAC regularizers and our policy regularization are in the same spirit, aiming to ensure candidate policies do not stray too far from the behavior policy, our Q-value regularization explicitly aims to mitigate extra-overestimation error which is not the case in BRAC. Specifically, BRAC's value penalty adds a divergence function between policy-determined distributions over actions (i.e. $\hat{D}(\pi_\theta(.|s'), \pi_b(.|s'))$) to the target Q-value which operates on the action-space (Here is Eq 6 from BRAC paper to make comparison easier):
>
> \begin{align*}
> \min_{Q_\psi} ~E{\substack{(s,a,r,s')\sim\mathcal{D}\\ a'\sim\pi_\theta(\cdot|s')}}{\left( r + \gamma\left(\bar{Q}(s', a') - \alpha \hat{D}\left(\pi_\theta(\cdot|s'), \pi_b(\cdot|s')\right) \right) - Q_\psi(s, a) \right)^2}~~~\text{Eq 6 from BRAC Paper}
> \end{align*}
>
> However, our CDC Q-value regularizer explicitly operates in Q-function space to prevent learning of poor Q-values in the presence of extra-overestimation stemming from the propagation of extrapolated temporal difference errors (compare our Q-value penalty below with the above BRAC equation):
>
> \begin{align*}
>  \min_{Q\theta}   E_{(s,a, s') \sim \mathcal{D}}  \left[ \Big( Q_{\theta}(s, a) - \mathcal{\overline{T}} Q_{\theta_{t-1}}(s,a) \Big)^2 + \eta  \cdot \Delta(s,a) \right]~~~~~~\text{Our Q-value regularizer}
> \end{align*}
>
> Note that our empirical results show that our method comfortably outperforms both versions of BRAC.  Even imposing strong policy constraints, the effectiveness of the resulting policy remains heavily affected by Q-value estimation errors, and thus our paper argues that it is imperative to regularize the Q-update in a way that specifically reduces these errors (unlike the  value penalty used in BRAC).
>
> > relation between the two regularizers proposed in the paper, are they independent components?
>
> Although our paper demonstrated that each regularizer can be used independently and their implementation is modular, we emphasize that they complement each other: the value-update mitigates extra-overestimation error while the policy regularizer ensures candidate policies do not stray too far from the offline data. Our ablation studies clearly show (see Figure 2a and Table S2) the best performance will be achieved only when both co-exist. We value the simplicity of our regularization terms: each penalty can be added to existing actor-critic RL frameworks with minimal extra code and the addition of both penalties involves no further complexity beyond the sum of the parts.
>
>
>
> > motivation of the policy regularizer. In BRAC dual format, the policy regularizer is also free of the estimation of behavior policy.
>
> Good observation, we will add more discussion of this point to the paper. Both BRAC’s and our policy regularization follow the same motivation that policy updates should favor not only actions with the highest estimated Q-value but also the actions observed in the dataset. Their difference comes from how to impose this type of constraint and how to optimize. As we emphasize in the paper, various f-divergences or Integral Probability Metrics could be employed for this purpose.  Our design philosophy in CDC is to utilize the simplest formulation which only requires the addition of a likelihood term to the existing policy objective.  We’ll clarify these motivations further in the discussion of Lemma 1.
>
>
>
> > one ablation study is needed to show the advantage of the policy regularizer in CDC. If one of the regularizers is replaced by the technique used in other offline RL algorithms, how will the overall performance change?
>
> Please see Figure 2a and Table S2 which show that both penalties are critical for the strong performance of our method, with the extra-overestimation penalty being of greater importance than policy regularization. We will clarify that we do not claim CDC’s policy regularizer to necessarily be the optimal choice for constraining the policy, but it is a particularly simple choice that combines very effectively with CDC’s Q-value penalty (see in particular results for environments like hopper/kitchen where neither policy nor value regularization alone performs very well). We find that our simple policy regularizer alone ($\eta=0$) performs similarly to BRAC-P and BRAC-V  (Tables S1 and S2) in many environments, yet all of these approaches that solely impose policy constraints are vastly outperformed by other batch RL methods. These results indicate that the use of combined policy and value regularization is far more important than which exact form of policy-regularizer is employed.

---

> > ### Comment · Reviewer_JguC · 2021-08-18
> > **Thank authors for the response**
> >
> > Thanks for the explanation of the differences between CDC and BRAC, the motivations are clear to me now.

---

> > > ### Author Response · Authors · 2021-08-18
> > > **Thanks.**
> > >
> > > Thanks again for your feedback and comments. We are glad that our responses address your concerns and we hope you will consider increasing your score. Please let us know if you have more questions.

---

### Official Review · Reviewer_XtLh · 2021-07-16

**Rating:** 6
**Confidence:** 5

**Summary:**

The authors propose Continuous Doubly Constrained (CDC) Batch RL. It adds two straightforward penalties, a policy constraint and a value-constraint. The policy constraint prevent the policy from taking actions that are not supported by the dataset, and the value-constraint prevents the value from being overly optimistic. Both penalties improve performance on the offline RL benchmarks compared to the baseline, but the two penalties combined work substantially better. They also perform competitively against several other offline RL baseline agents (BC, BCQ, BEAR, BRAC-V, BRAC-P, CQL).

**Ethical Concerns:**

I have no ethical concerns with this work.

**Limitations And Societal Impact:**

The limitations and societal impact are adequately addressed.

**Main Review:**

## Originality:
The method is a novel combination of well-known techniques. Specifically, it combines policy-constraint offline RL methods and value-constraint offline RL methods. Both have been prevalent in the literature, and this paper shows that there is a clear advantage to combining both approaches. It is clear to me how this method differs from previous contributions, and related work is adequately cited.

## Quality:
Overall I think the paper is technically sound, and its main claims are well supported.
It shows through experimental results that combining policy-constraints and value-constraints has a clear advantage across a well established benchmark for offline RL. And shows that combined constraints outperforms a number of baseline agents.

The main weakness in my mind is the authors introduce a new policy-constraint and a new value-constraint, but it is not clear the policy-constraint is better than existing policy-constraint methods e.g. BCQ, BRAC, or that the value-constraint is better than existing value-constraint methods e.g. CQL.

Indeed the best PC method BCQ gets 1060, while theirs gets 299, and the best VC method CQL gets 1245, while theirs gets 555.

Is that correct? I am comparing results across tables in the appendix so I am not sure if my comparisons are fair.

Can the authors explain why they think their PC and VC methods perform so much worse than the baselines? I think it may be interesting to make these comparisons in the main body of the paper, and clarify the discrepancies.

## Clarity:
Overall I think the paper is clearly written and well organized. And I think an expert reader would be able to reproduce its results.

## Significance:
I think the results are important. Offline RL is an important research direction and these results show that policy and value constraints can be combined to get better performing algorithms. In that sense, it addresses a difficult task in a better way than previous work.

## Additional Questions:
On line 79 you mention "In particular, we do not even require that complete episode trajectories have been logged [59, 62]."
Can you clarify why CRR and BRAC require complete episode trajectories?

In Section 5.1 you show results for Offline Policy Evaluation. Your FQE baseline on the D4RL tasks performs quite poorly in comparison to the FQE baseline in [this benchmark paper](https://arxiv.org/abs/2103.16596), which also uses policies trained on the D4RL datasets. Do you have any idea what might account for this difference?

**Time Spent Reviewing:**

5

---

> ### Author Response · Authors · 2021-08-10
> **Response to Reviewer XtLh**
>
> ​​Thank you for your valuable feedback.
>
> >  authors introduce a new policy-constraint and a new value-constraint, but it is not clear the policy-constraint is better than existing policy-constraint methods e.g. BCQ, BRAC, or that the value-constraint is better than existing value-constraint methods e.g. CQL.
>
> The main goal of this work is to propose a new and simple batch RL approach that builds on top of the standard off-policy actor-critic RL methods by adding a simple pair of regularizers without significant changes to the underlying algorithm. In contrast, the policy constraint in method like BCQ is based on a variational autoencoder framework that requires encoder-decoder networks, a new objective function involving stochasticity (i.e. reconstruction loss + KL term which requires sampling), and thus changes to inputs and forward-pass computation of the policy network. On the other hand, our simple likelihood penalty to regularize the CDC policy is easily compatible with arbitrary models and without the more complex changes required in order to use BCQ. Note that even with a strong policy constraint, the resulting policy is still affected by the learned Q-value, so it remains critical that we still correct Q-value issues as well.
>
> CDC's $\Delta$ penalty differs from the CQL penalty because CQL penalizes the Q-value for all proposed actions (its goal is overall pessimism, ie. to lower-bound the Q-value), whereas CDC only penalizes Q-values for proposed actions that were previously unobserved with large (extrapolated) values. CDC only aims to reduce these values toward the values assigned to the best previously observed actions in the dataset, unlike CQL which pushes all Q-values down (CDC only aims for the minimal reduction in Q-values needed to mitigate extra-overestimation, not for lower-bounding the true Q-values like CQL).
>
>
> We have added to the Discussion section that:
>
> “Unlike previous work, this paper highlighted the importance of simultaneous policy and value regularization in batch RL. One can envision other combinations of alternative policy and value regularizers that may perform even better than the particular policy/value penalties used in CDC. That said, our CDC penalties are particularly simple to incorporate into arbitrary actor-critic RL frameworks, and operate synergistically as illustrated in the ablation study from Figure 2a and Table S2 (where for environments like hopper/kitchen, the performance boost from simultaneous application of both penalties far exceeds the individual gains from either one)”.
>
> Please also see our response to Reviewer JguC about detailed comparison with BRAC.
>
>
> > On line 79 you mention "In particular, we do not even require that complete episode trajectories have been logged [59, 62]." Can you clarify why CRR and BRAC require complete episode trajectories?
>
> Thanks for highlighting this point. We clarify that neither CRR nor BRAC requires complete episode trajectories and both work with just a single-step transition. The intention of that poorly worded sentence was to emphasize the fact that our method, like CRR and BRAC, only needs a dataset consisting of a single-step transitions and does not require complete episode trajectories. We will clarify this in the final version.
>
> On a slightly related note, the CRR paper [59] discusses a potential problem with using k-step trajectories for advantage estimation in batch RL, demonstrating that longer trajectories can potentially lead to “risk seeking” or otherwise “undesirable behavior”. Please see sections “A.3 Effects of using K-step returns” and “CRR vs. return-based methods” in the CRR paper for more details.
>
>
> > In Section 5.1 you show results for Offline Policy Evaluation. Your FQE baseline on the D4RL tasks performs quite poorly in comparison to the FQE baseline in this benchmark paper, which also uses policies trained on the D4RL datasets. Do you have any idea what might account for this difference?
>
> Thanks for the nice reference, which we’ve added to our paper. Our OPE experiment setup can be roughly compared to FQE-L2 under the rank correlation metric in the DOPE-Benchmark paper (Fu et al., 2021). The DOPE paper reveals that importance sampling and doubly-robust policy evaluation outperform FQE-L2 under the rank correlation metric for OPE in the D4RL benchmark (see Figure 5 and page 8 in DOPE paper). While our experiment didn’t follow their exact setting and there are differences in the details, both papers come to the same conclusion that FQE-L2 under the rank correlation metric is not good enough for effective OPE on a challenging benchmark like D4RL. This is in line with our subsequent result that shows FQE with our penalty performs substantially better than standard FQE (i.e. FQE-L2).

---

> > ### Comment · Reviewer_XtLh · 2021-08-25
> > **Thank you for your response.**
> >
> > Thank you for your response. The text you have added to the Discussion section is helpful.
> >
> > **Regarding OPE:** Your results agree with that paper, in the sense that FQE-L2 doesn't perform well on D4RL. But in your setup FQE-L2 performs even worse (average correlation near 0, high negative correlation for many tasks). Do you have any idea what about your setup might account for this difference? I'm just curious.
> >
> > Nevertheless, it seems like your proposed method improves things.

---

> > > ### Author Response · Authors · 2021-08-27
> > > **Thanks.**
> > >
> > > Thanks again for your feedback and comments. We are glad that you found our responses helpful. We hope you will consider increasing your score.
> > >
> > > > Regarding OPE:
> > >
> > > The source of variation in the candidate policies in DOPE, as explained in their section 4, is extremely different from the candidate policies in our paper (we simply utilized various versions of Algorithm 1 with various hyperparameters to generate our candidate policies). The ability of an OPE method to identify the better policies from the candidates will thus inevitably differ in the experiments in this paper vs DOPE. In addition, normalization of (real) returns and estimated returns can be another source of variation where DOPE-Benchmark normalized those to be between 0 and 1 while it was not the case in our paper. Nonetheless, despite all those differences, both papers have reached the same qualitative conclusions about FQE-L2. We will further clarify our setup in the paper.
> > >
> > > Please let us know if you have more questions.

---

### Author Response · Authors · 2021-08-10
**Response to all reviewers**

​​We thank the reviewers for their feedback that we've used to greatly improve the paper. We have responded to the concerns of the reviewers as individual comments below.

All reviewers agree that our experimental results are solid, main claims are well supported, and our contributions are valuable and important to the community. To summarize our contributions: (i) we propose a simple pair of regularizers that are effective for batch RL and demonstrate that  the use of both policy and value regularization outperforms existing methods in large-scale batch-RL benchmarks. (ii) In addition to our comprehensive results, our theoretical results provide further insights as to why our method performs favorably.

Per Reviewer T9dA’s suggestion, we also added another (recently published) baseline to further situate our method relative to the literature. This result provides another data point that our method is a practical and effective approach for batch RL. Please see our response to Reviewer T9dA where we discuss this new experiment.

---

> ### Comment · Program_Chairs · 2021-08-20
> **This comment is not visible to the Area Chair handling your paper**
>
> Dear Authors,
>
> Please edit the Readers field to include the Area Chair handling your paper.
>
> Thanks,
> Program Chairs

---

> > ### Author Response · Authors · 2021-08-20
> > **Updated Readers field to include the Area Chairs**
> >
> > Dear Program Chairs
> >
> > Our apologies, that was a mistake on our side.  You are right, we forgot to include "Paper3048 Area Chairs" in our "Response to all reviewers" comment. We edited the Readers field and now included the area chairs. All our comments and responses now include the following readers:
> >
> > Program Chairs, Paper3048 Senior Area Chairs, Paper3048 Area Chairs, Paper3048 Reviewers Submitted, Paper3048 Authors
> >
> > Please let us know if there are any visibility issues with our responses and comments.
> >
> > Thanks,
> >
> > Authors

---

> ### Author Response · Authors · 2021-08-20
> **Updated Readers field to include the Area Chairs**
>
> Dear Area Chairs,
>
> It was brought to our attention by Program Chairs that "Paper3048 Area Chairs" were not included in this response. We apologize for that.
> We now updated the Readers field to include  "Paper3048 Area Chairs".
>
> Please let us know if you have any visibility issues with our responses and comments to all reviewers.
>
> Thanks,
> Authors

---

> > ### Comment · Area_Chair_QN2o · 2021-08-23
> > **Thank you**
> >
> > Thank you for resolving this, the visibility is now correct.

---

### Decision · Program_Chairs · 2021-09-28

**Decision:**

Accept (Poster)

**Comment:**

The authors have proposed an algorithm for (offline) batch RL that leverages a pair of regularisers, one constraining the policy and the other constraining that value, that lead to improved performance across a broad set of continuous RL benchmarks.

The reviewers agree that the results are convincing, well-supported by theoretical analysis and overall clearly presented. There is some discussion whether the individual regularisers themselves are especially novel w.r.t. prior work, but they are evidently complementary and yield convincing empirical performance. The authors have gone to noteworthy lengths to address all reviewer feedback, including running new ablations (yielding consistent results) and expanding their literature review to capture methods published around the submission deadline.

I fail to identify any reviewer concerns that have not been sufficiently addressed by the authors' detailed responses, and I agree with the reviewers' assessment that this is a high-quality submission well-suited for publication at NeurIPS.

We thank the authors for their submission and effort engaging with the review process.


**Consistency Experiment:**

NeurIPS has a long history of experimentation. In 2014, NeurIPS ran an experiment in which 10% of submissions were reviewed by two independent committees to quantify the randomness in the review process. This year, we repeated a variant of this experiment to see how the quality of the review process has changed over time.  This paper was part of the experiment and was therefore assigned to two committees (consisting of reviewers, an Area Chair, and a Senior Area Chair) that reached independent decisions.  If both committees made the same recommendation, this recommendation was followed. If a single committee recommended acceptance, the paper was accepted (with the exception of a few cases in which the other committee identified what we considered a fatal flaw, e.g., an error in a key result).

This copy’s committee reached the following decision: **Accept (Poster)**

The other committee assigned to the paper recommended **Reject**.  You can find the other set of reviews, along with any follow up discussion with the authors here:
https://openreview.net/forum?id=RIEqVBFDJTR